# Carbon Electrodes: The Rising Star for PSC Commercialization

**Maria Bidikoudi * and Elias Stathatos ***

Department of Electrical and Computer Engineering, University of the Peloponnese, 26334 Patras, Greece
* Correspondence: mbidikoudi@uop.gr (M.B.); estathatos@uop.gr (E.S.)

**Abstract:** After more than 10 years of intensive optimization, perovskite solar cells (PSCs) have now reached the point where the step towards their commercialization is expected. In order to move in this direction, the upscaling of devices is mandatory. However, the metal electrodes employed in the highest performing PSCs constitute a major obstacle, being both costly and unstable. In this review, the replacement of metal electrodes with carbon (C) electrodes in high-performing perovskite solar modules (PSMs) is presented. An overview of the background and current status is addressed, the potential of this material is highlighted and the challenges and future prospects are discussed.

**Keywords:** perovskite solar cells; large area; carbon electrode; carbon perovskite solar cells; perovskite solar modules

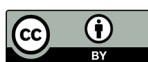

## 1. Introduction

It has been more than a decade since perovskite structures were successfully applied as absorbers in thin-film solar cells [1], and since then, a huge leap in power conversion efficiency (PCE) has been achieved, stemming from extensive and intensive research in this field. After having established PCEs that exceed 25% in lab-scale devices, research is now turning to upscaling, which is the next step towards the commercialization of this technology, which can be considered mature enough by now. The advances in large-area perovskite solar cells (PSCs), namely perovskite solar modules (PSMs), have yielded PCEs close to 18% [2], with adequate operational stability. Owing to this development, there is now an increasing number of companies worldwide that fabricate PSCs with the scope of introducing them to the market. However, the challenges of successfully turning this technology from a "lab success" to a viable product are plenty.

A typical, high-performing perovskite-based solar-harvesting device has the conductive substrate/CTL/perovskite/CTL/metal electrode multilayer configuration, where CTL refers to the charge transport layer, i.e., the electron transport layer (ETL) and hole transport layer (HTL), and the metal electrodes are most commonly gold (Au) and silver (Ag). The use of metal electrodes is one major obstacle to be tackled for the upscaling of PSCs. Besides the physical and chemical limitations, including ion migration and corrosion, which compromise the stability of the devices, the technical and financial limitations are not negligible. Metal electrodes require high-vacuum techniques for their deposition, which in turn require specialized equipment and inert conditions. The electrode deposition on large-area substrates, therefore, results in a demanding and costly process, which elevates the cost of the resulting PSMs.

A different type of PSC architecture has been receiving an increasing amount of attention the past 5 years, where metal electrodes are replaced by carbon (C) electrodes, typically prepared by a C slurry. These devices are favored by a number of advantages compared to the typical PSCs. The C electrodes are prepared by low-cost starting materials, using simple, scalable and industrially compatible methods. Moreover, the resulting

devices, namely C-based PSCs (C-PSCs), can be fabricated in ambient conditions, can be HTL-free and are distinguished for their outstanding stability. These properties have led the scientific community to turn to this device architecture, in order to proceed to the up-scaling of PSCs.

The aim of this manuscript is to highlight the potential of carbon electrodes as the back contact of low-cost and stable perovskite solar modules. For this purpose, in the first part of this manuscript, a retrospect is made in the successful application of C electrodes as counter electrodes (CE) in a variety of next-generation solar cells, i.e., dye- and quantum-dot-sensitized solar cells (DSSCs, QD-SSCs) and organic solar cells (OSCs). In the second part, an introduction to perovskite solar cells is made, and the working principle and device architectures are presented, with a focus on PSCs with a C electrode, where the examples of the highest performing devices are summarized. The third part of the manuscript targets the application of carbon electrodes in perovskite solar cells of large area, i.e., perovskite solar modules, and is the main part of this work. The fabrication methods of carbon electrode perovskite solar modules (C-PSMs) are presented, together with the interconnection methods, and an in-depth literature review is performed to describe the current status of C-PSMs and the state of the art. A cost analysis of C-PSMs is presented to provide an overview of the economic aspect of turning to carbon electrodes for perovskite solar cell upscaling. Finally, the challenges and future prospects of C-PSMs are briefly discussed.

With this review, we intend to provide the readers with the most up-to-date developments in the field of PSMs and the commercialization prospects of PSMs with C electrodes, which are currently in the spotlight of industry-oriented research, and which we believe will be the technology of the future in the solar cell field.

## 2. Background

### 2.1. Dye-Sensitized Solar Cells (DSSCs)

Dye-sensitized solar cells (DSSCs) are a third-generation photoelectrochemical type of solar cell, which in principle mimic the natural process of photosynthesis to produce electricity, through a photogenerated potential difference. These cells were originally co-invented in 1988 by Brian O'Regan and Michael Grätzel and were further developed until 1991, when PCE of 7% was obtained and the potential of this technology arose [3]. Since then, extended research and optimization of all the individual parts of DSSCs has yielded certified PCE of 13% in 2020 [4], while impressive PCE reaching 34% has been achieved in low light illumination conditions (ambient lighting) [5]. A DSSC consists of a dye-sensitized mesoporous oxide (typically TiO$_2$) working electrode (anode), a redox electrolyte and a counter electrode (cathode), which has the role of catalyzing the reduction process of the electrolyte's redox couple and at the same time serves as the hole collector (Figure 1). DSSCs are considered the "ancestors" of mesoscopic perovskite solar cells, since perovskites were initially used as sensitizers in solid-state DSSCs.

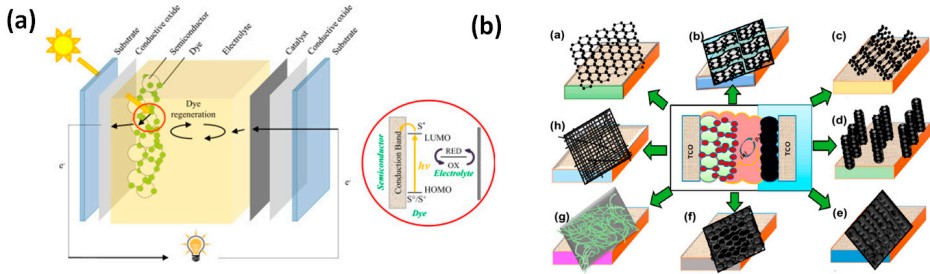

**Figure 1.** (**a**) A sketch of the DSSC structure and operational mechanism [6]. (**b**) Schematic diagrams showing different types of carbon materials used as CEs in DSSCs (in the subfigure (a) graphene, (b) graphite, (c) randomly oriented carbon nanotubes, (d) one-directional carbon nanotubes, (e)

carbon black, (f) activated carbon, (g) carbon nanofibers, and (h) hollow active carbon nanofibers) [7]. (Reproduced from Ref. [6], 10.1016/j.rser.2014.07.079, under the terms of the CC BY [4.0] license, and Ref. [7], 10.3390/ma13122779, with permission).

A variety of carbon structures have been employed as counter electrode materials to replace the originally used platinum (Pt) in DSSCs, with great success. These materials have attracted attention because of their low cost, stability and high catalytic ability towards various redox couples, including cobalt (Co) and copper (Cu), for which Pt is not suitable. Some of them include carbon black (CB) [8], carbon nanotubes (CNTs) [9], carbon nanotube fibers (CNFs) [10,11], carbon nanohorns (CNHs) [12,13] and mesoporous carbon [14]. In fact, PCEs that exceed 14% have been achieved with novel carbonaceous CEs, outclassing the efficiency records that have been obtained with the most widely used Pt electrodes [15].

Graphitic–carbon black mixture was the first carbon-based CE developed by Gratzel in 1996 with photoconversion efficiency of 6.7% [16]. By tailoring the particle size and thickness of the electrode, Murakami et al. have achieved PCE of 9.1% using carbon black as the carbon electrode material [17]. Benefiting from its mesoporous structure which results in many active sites for the redox couple reduction, Zhao et al. have prepared carbon aerogel CEs that were incorporated in flexible DSSCs that have achieved PCE as high as 9.06% [18].

Lodermeyer et al. reported on a novel carbon allotrope, namely single-walled carbon nanohorns (SWCNH), incorporated into devices that yielded efficiencies as high as 7.7% when used as a CE material in DSSCs with the $I^-/I^{3-}$ redox couple and a Ru-based dye [12].

An interesting approach has also been proposed by Monreal-Bernal et al., where free-standing carbon nanotube fibers (FSCNFs) have been used as dual-function CEs and current collectors, leading to η as high as 8.8% and stabilities of ca. 70% after 6 months; additionally, large-area DSSCs were prepared, employing a 1 m long CNTf-CE, which exhibited average η comparable to the state-of-the-art carbon-based DSSC modules [19].

A Co-redox couple-based electrolyte has been paired with thiocyanate-free isoquinazolylpyrazolate Ru-(II) complexes and a thermally deposited carbon counter electrode in DSSCs that resulted in 9.53% PCE [20].

Counter electrodes that comprise copolymer-templated nitrogen-enriched nanocarbons (CTNCs) have been used in DSSCs employing the $Co^{2+/3+}$ redox couple and obtained PCE up to 10.32%, which was attributed to the high surface area of CTNCs and to their superior electronic properties that originate from the nitrogen heteroatoms on the edges of nanographitic domains [21].

Using the same redox couple, Zhu et al. prepared CEs based on carbon spheres. After optimizing the diameter, hollow structure and open-ended surface, properties that determine the catalytic behavior of the carbon spheres, they were able to obtain DSSCs with a high PCE of 10.61% [22]. A higher PCE of 10.71% has also been obtained by modifying the carbon spheres and in particular by preparing a core–shell structured paramagnetic carbon sphere with carbon as shell and paramagnetic $Fe_3O_4$ as core, while the composite material has been additionally been deposited on magnetic substrate [23].

In 2017, Kim et al. prepared activated anchovy-336-derived carbons (AnCs) which were employed as CEs in DSSCs with a $Co(bpy)_3^{2+/3+}$ redox mediator and a porphyrin sensitizer and achieved a PCE of 12.72% and superior electrocatalytic activity compared to Pt [23].

In the iodine-free electrolyte context, a Cu-based electrolyte has been incorporated in DSSCs with an SWCNT CE, which exhibited PCE of 7.5% at 1 sun illumination, while under low light conditions (0.5 sun illumination) it further rose to 8.3%, suggesting a potential use for indoor applications [24].

Finally, a highly promising monolithic DSSC device configuration has been recently proposed by Kokkonen et al. [25], in which the CE has been replaced by a printed layer of

CNTs, in a device that is fully printable and pushes DSSCs closer to the integration of large areas (Figure 2).

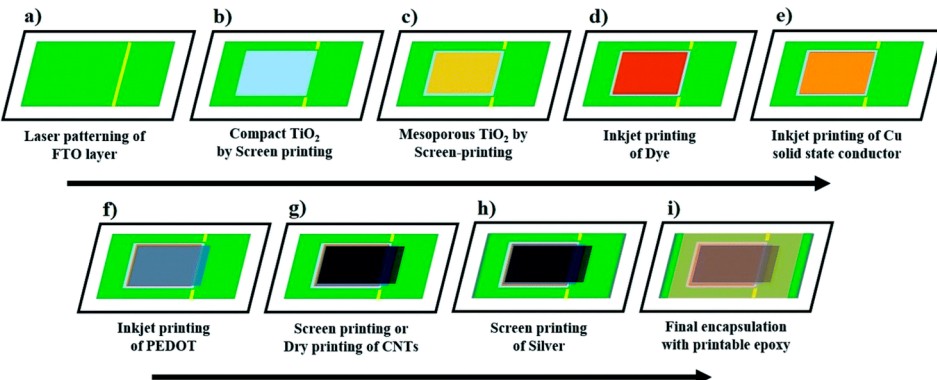

**Figure 2.** Proposed process flow for producing advanced monolithic DSSCs with Cu redox shuttle-based electrolytes and solid hole conductors [25]. (Reproduced from Ref. [25] 10.1039/D1TA00690H with permission from the Royal Society of Chemistry, under the terms of the CC BY [3.0] license.)

The aforementioned results are only a few examples of the most efficient devices that have been reported, where carbonaceous materials are used as a replacement of Pt in counter electrodes. However, these examples bring out the versatility of C and the potential that it presents as a material for universal and multifunctional electrodes, with the capability to be incorporated in highly efficient DSSCs with a variety of redox couples and sensitizers.

### 2.2. Quantum-Dot-Sensitized Solar Cells (QD-SSCs)

Quantum-dot-sensitized solar cells (QD-SSCs) are a similar photoelectrochemical type of solar cell to DSSCs, with the differentiation of the absorber material, which in this type of device consists of semiconducting quantum dots (e.g., PbS, CdS and CdSe), instead of dye molecules. These materials are characterized by some desirable properties, including an easily tunable band gap through size and composition control; a high molar extinction coefficient; light, thermal and moisture stability; and, most interestingly, multiple exciton generation (MEG) capability, which can lead to QD-SSCs surpassing the Shockley–Quessier efficiency limit of 32.9% (for single absorber solar cells) [26,27]. Similarly to DSSCs, DQ-SSCs comprise an electron transport layer, which typically is a wide bandgap semiconductor, an excitonic quantum dot (QD) sensitizer, a polysulfide redox electrolyte as the hole transport media and a counter electrode. In QD-SSCs, the CE materials that are most commonly used are metal sulfides and conducting polymers. Since their first report in 2002, QD-SSCs have significantly risen the obtained PCE, as seen in Figure 3, from ~0.5% [28] to 18.1% in 2020 [4]. The replacement of metal sulfide catalysts, such as $Cu_2S$, which has been widely used as a CE material, with carbon has been a determinant of the tremendous improvement in the devices' performance from 2016 and on. Up to this point, QD-SSCs were limited to obtaining PCEs of no more than 10%.

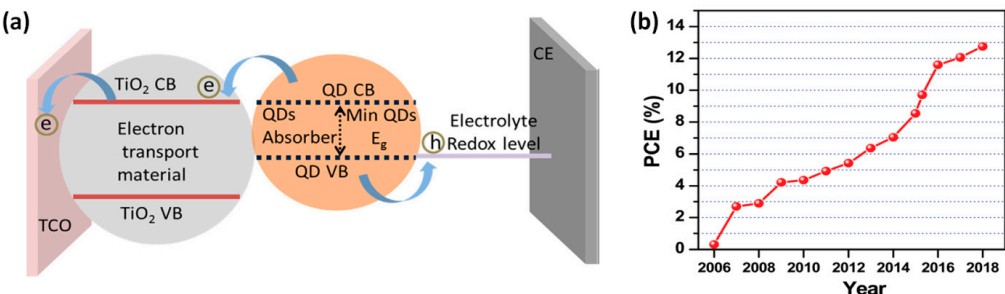

**Figure 3.** (**a**) Schematic diagram of quantum-dot-sensitized solar cells (QD-SSCs) [29]. (**b**) Evolution of the record PCE of QD-SSCs since 2006 (for cells based on a standard two-electrode configuration and tested under the irradiation of air mass (AM) 1.5 G one full sun illumination) [30]. (Reproduced from Ref. [29], 10.1016/j.solener.2020.04.044, and Ref. [30], 10.1039/C8CS00431E, respectively, with permission).

The most successful method to employ C in counter electrodes for application in QD-SSCs was presented by Du et al. [31] in 2016. In their work, the authors describe the fabrication and characterization of a CE that consists of mesoporous carbon (MC) paste deposited and calcinated on a titanium (Ti) mesh substrate, instead of the typical fluorine-doped tin oxide (FTO) glass (MC/Ti). With this combination of materials, the favorable properties of the metal substrate, such as high conductivity, low sheet resistance and superior chemical stability, are combined with optimum mechanical adhesion with the C paste (Figure 4). The result has been a high-performing C-based electrode that has been widely used since then, resulting in devices which exhibit significantly enhanced PCE values (Table 1).

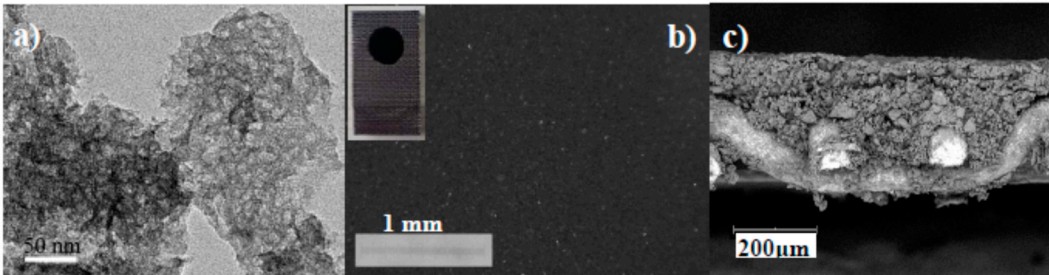

**Figure 4.** MC material and derivative MC/Ti CEs. (**a**) Representative transmission electron microscopy (TEM) image, (**b**) top view image of the front side of a MC-4/Ti CE with the photograph in the inset, (**c**) scanning electron microscope (SEM) image of cross-section of MC-4/Ti CE [31]. (Reproduced from Ref. [31], 10.1021/acs.jpclett.6b01356, with permission.)

To mention a few, Ti mesh-supported mesoporous carbon electrodes have been used as CE, were combined with Zn–Cu–In–Se (ZCISe) alloyed QDs, which are both Pb- and Cd-free, and achieved certified PCE of 11.61% (champion 11.91%) [32]. The same group further modified the mesoporous carbon with nitrogen, obtaining certified PCE of 12.07% (champion 12.45%), suggesting that further modifications could lead to devices of very high performance [33]. One of the highest PCEs of QD-SSCs has been obtained with the use of MC/Ti electrode in devices that employ Cu-deficient Zn–Cu–In–Se (ZCISe) QDs, as presented by Zhang et al. [34]. In their work, the authors present a champion PCE of 12.57% ($Jsc$ = 25.97 mA/cm$^2$, $Voc$ = 0.752 V, FF = 0.644) that was achieved by employing ZCISe QDs with Cu/In molar ratio of 0.7 under AM 1.5 G one full sun illumination. The same CE has been used by Z. Du et al. in the initial work presenting the MC/Ti electrode, combined with $CdSe_{0.65}Te_{0.35}$ QDs, which yielded QD-SSCs with certified PCE as high as 11.16%, demonstrating the functionality of mesoporous carbon electrodes in various types of QDs as absorbers and highlighting the versatility and superior catalytic activity of this material [31].

In 2018, Wang et al. boosted the performance of ZCISe QD-sensitized, mesoporous carbon (MC/Ti) CE solar cells by co-sensitizing the anode ($TiO_2$) with CdSe quantum dots (QDs) and in combination with a polysulfide electrolyte with the poly(vinyl pyrrolidone) (PVP) additive, presenting PCE of 12.75% and a highly impressive generated photocurrent density value of 27.39 mA/cm$^2$ [35]. Finally, the MC/Ti electrode has also been successfully incorporated in solar cells employing the $Cu–In_{0.7}–Ga_{0.3}–Se$ (CIGSe) QDs that, in combination with a typical polysulfide electrolyte, exhibited champion efficiency of 11.49% [36].

**Table 1.** Summary of the highest performing QD-SSCs reported that use C-based electrodes.

| CE Material | Type of QD | Electrolyte | PCE | Year | Ref. |
|---|---|---|---|---|---|
| **MC/Ti** | $CdSe_{0.65}Te_{0.35}$ | polysulfide | 11.16 | 2016 | [31] |
| **MC/Ti** | Zn–Cu–In–Se | polysulfide | 11.61 | 2016 | [32] |
| **N-doped MC/Ti** | Zn–Cu–In–Se | polysulfide | 12.07 | 2017 | [33] |
| **MC/Ti** | Zn–Cu–In–Se | polysulfide | 12.57 | 2017 | [34] |
| **MC/Ti** | Zn-Cu-In-Se/CdSe | polysulfide + PVP | 12.75 | 2018 | [35] |
| **MC/Ti** | $Cu–In_{0.7}–Ga_{0.3}–Se$ | polysulfide | 11.49 | 2017 | [36] |

*2.3. Organic Photovoltaics (OPVs)*

Organic solar cells (OSCs) or organic photovoltaics (OPVs) are third-generation solar cells that use organic polymer materials as the light-absorbing layer. Even though the working principle is the same as all emerging solar cell technologies, the device architecture diverges. In brief, in this case, there are two different organic materials that are being used, the electron donor (p-type semiconductor) and electron acceptor (n-type semiconductor), which are "sandwiched" between the anode and cathode of the device. These can be incorporated as discrete thin layers (planar heterojunction structure (PHJ)) or in a single blend which is cast as a mixture (bulk heterojunction (BHJ)) (Figure 5).

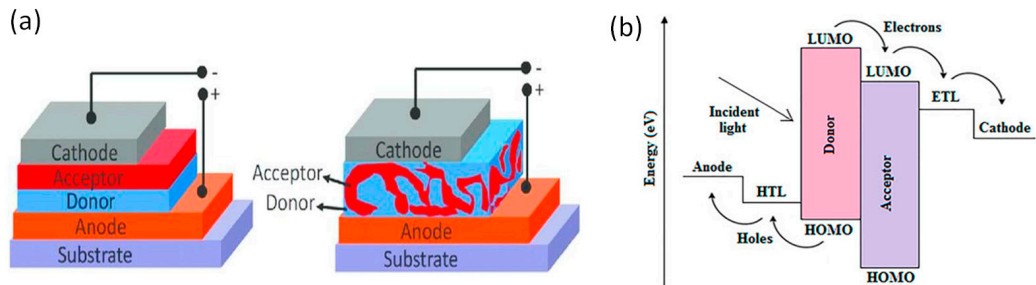

**Figure 5.** (**a**) Planar and bulk heterojunction organic solar cells [37]. (Reproduced from Ref. [37],10.21533/pen.v5i2.105, under the terms of the CC BY [4.0] license.) (**b**) Schematic diagram of the band structure of an organic solar cell [38]. (Reproduced from Ref. [38], 10.3389/fchem.2021.733552, under the terms of the CC BY [4.0] license.)

They were first reported in the 1980s when the first polymers, such as poly(sulfur nitride) and polyacetylene, were implemented in solar cells, delivering very low PCEs. It was in 1986 when Tang et al. reported PCE as high as 1%, by bringing a donor and an acceptor together in one cell [39], that the research interest started to turn to this type of solar cell. Since then, a tremendous increase in PCE values has been achieved, after enormous efforts in materials design and device engineering, and at this moment, certified PCE of 19.2% has been achieved [40]. Organic solar cells are the most advanced among the next-generation solar cells in terms of commercialization, owing to their key features including the organic semiconductors' ability to be tailored, their being based on abundant and non-toxic raw materials, and that their fabrication methods (e.g., vacuum coating, solution processing) are capable of transferring to the large area with coating and printing methods including slot-die coating, screen and inkjet printing, and the roll-to-roll process, under ambient conditions. Moreover, they have low material consumption, low temperature processing and the ability to be casted onto flexible substrates [41].

The most widely used carbonaceous materials that have been used so far in OSCs are graphene [42] and carbon nanotubes [38], both of them incorporated mostly as transparent electrodes, but also as active layers and interface layers with great success [43]. In particular, the need to increase the stability of these devices has led to the replacement of metal electrodes with a variety of alternatives. However, pristine carbon has been scarcely

employed in this type of solar cell, delivering very low PCE devices, suggesting that the transition from the aluminum (Al) cathode to C still needs a lot of improvement. The first report of metal electrode replacement with C was by Bennatto et al. in 2014 [44], where the authors presented an OPV module produced with the roll-to-roll manufacturing process that is indium tin oxide (ITO)-free and metal-free, and their work demonstrated printed carbon electrodes for consumer electronics with low environmental impact (Figure 6). Their devices exhibited an average PCE of 1.5% and their work showed that C as electrode material significantly lowers the manufacturing cost of OPV modules while retaining the flexibility, the active area efficiency, and stability.

Krebs et al. [45] have launched an initiative where ITO-free OPVs that are prepared by the roll-to-roll processing technology, laser cut and encapsulated are available free of charge. This initiative is called "freeOPV" and is aimed at the creation of a platform for the evaluation of processing technology by using information retrieved by researchers worldwide.

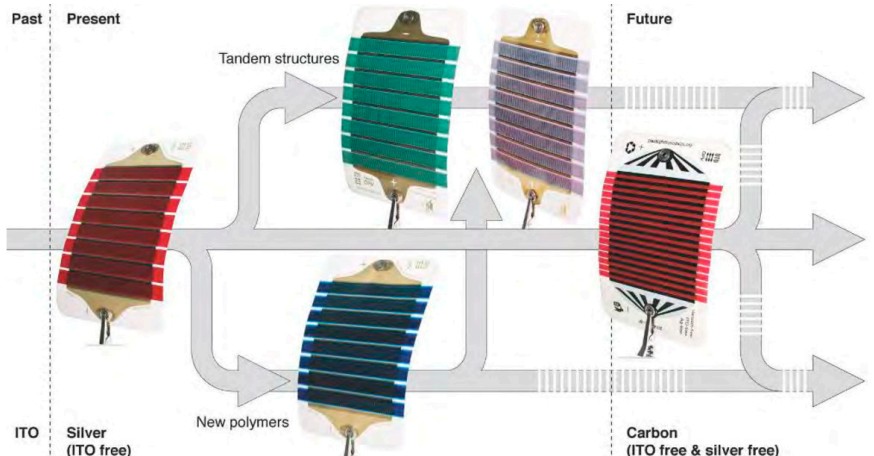

**Figure 6.** An illustration of the evolution of the freeOPV platform from its initial form based on slanted silver grids and poly(3-hexylthiophene):phenyl-C61-butyric acid methyl ester (P3HT:PCBM) as the active materials to encompass novel polymer active materials and present complex multilayer architectures such as the tandem freeOPV. Printed carbon electrodes for consumer electronics with low environmental impact are demonstrated [44]. (Reproduced from Ref. [44], 10.1002/aenm.201400732, with permission.)

Following the example of Bennatto et al., Perulli et al. [46] recently presented a module (10 × 14.2 cm²) that consists of 16 single cells that are connected in series with an estimated total active area of about 35 cm², where the Al cathode was replaced with C (Figure 7). The generated power output of the device showed a maximum value of power output (Pm) of 23.8 mW, corresponding to a power conversion efficiency (PCE) of 0.68%. The low performance was attributed by the authors to the lack of uniformity of cell efficiency within each module.

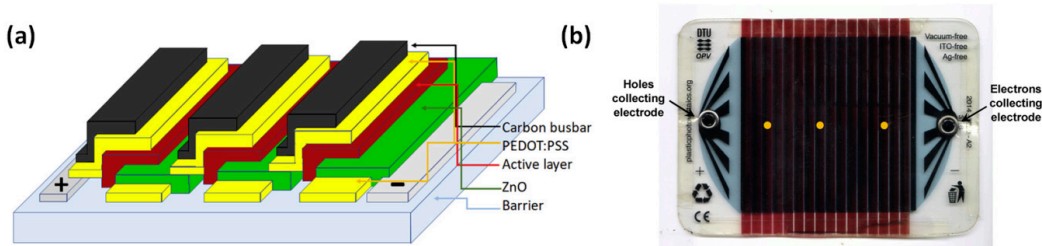

**Figure 7.** (**a**) Layer structure of the investigated photovoltaic module. (**b**) Photograph of the freeOPV module [46]. (Reproduced from Ref. [46], 10.1016/j.synthmet.2018.12.015, with permission.)

### 3. Application of C Electrodes in PSCs

*3.1. Perovskite Solar Cells (PSCs)*

In general terms, perovskite solar cells are next-generation solar cells, where materials with the perovskite structure are employed as the photoactive material (absorber). Perovskites are materials with the $ABX_3$ structure, where A and B are cations and X is an anion bonding to both, that have been under investigation for optoelectronics applications since the late 1990s, when the structural dynamics of methylammonium lead iodide perovskite ($MAPbI_3$) material were revealed [47]. The typical perovskite materials that are being used in high-performing PSCs are the hybrid organic–inorganic type (HOIP), where the A site cation is methylammonium (i.e., $MA(CH_3NH_3)$), formamidinium (i.e., $FA(CH(NH_2)_2)$) or a combination of both; the B site cation is mainly Pb; and X is typically a halogen-based anion. These perovskites have shown outstanding light-harvesting properties and to date, the highest power conversion efficiency (PCE) of 25.8% has been achieved in PSCs employing two FA cation perovskites, by coupling chlorine-bonded $SnO_2$ with perovskite precursors containing chlorine [48].

Depending on their structure, PSCs are divided in two main categories: the n-i-p or otherwise "normal" structure, and the p-i-n or otherwise "inverted" structure. The working principle and physical processes that take place in all device architectures are the same: light absorption by the photoactive material, charge generation, charge transport in the perovskite and charge extraction (with simultaneous charge recombination) by the selective contacts. The differences between the structures are mainly in the materials that are being used for each component.

In the n-i-p structure, light passes through the working electrode, which consists of an n-type material deposited on a transparent electrode (electron transport layer), on top of which the perovskite crystals (intrinsic semiconductor) are grown. The p-type material lies on top of the perovskite absorber (hole transport layer) and a metal contact completes the electrical circuit. Depending on the nature of the ETL, this type of device is subcategorized into mesoporous or planar. In the p-i-n structure, illumination occurs from the p-type electrode, where the hole transport material is deposited on a transparent substrate, forming the HTL. The perovskite is grown on the HTL and the deposition of the n-type material follows, forming the ETL. Similarly to the n-i-p structure, the deposition of a metal electrode completes the electrical circuit (Figure 8).

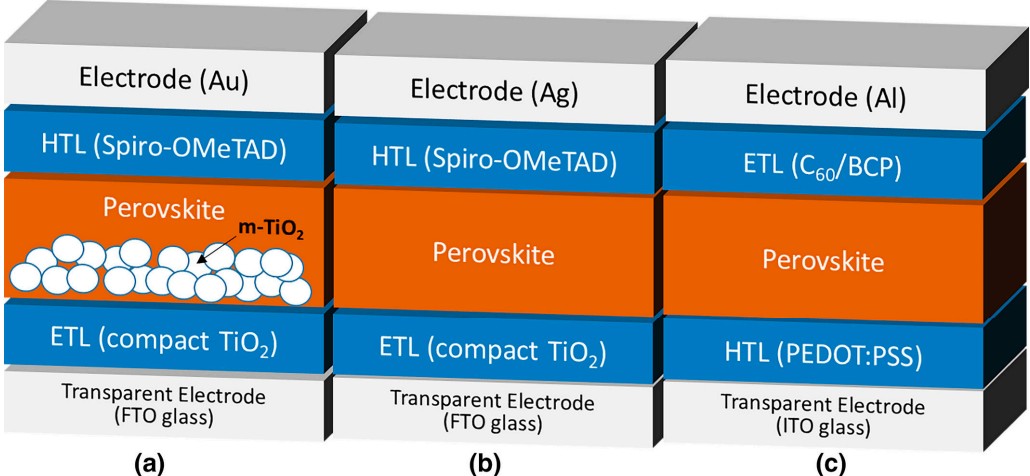

**Figure 8.** Device architectures for (**a**) mesoporous perovskite solar cell of the normal structure, (**b**) normal structure planar heterojunction perovskite solar cell and (**c**) inverted planar heterojunction perovskite solar cell [49]. (Reproduced from Ref. [49], 10.1002/eem2.12035, with permission.)

The advances in perovskite solar cell through the years have been substantial, both in terms of material design and device optimization. Among the parts of PSCs, the one

that has attracted a significant and increasing amount of attention is the back (counter) electrode. In particular, it has now been acknowledged that in order to progress and up-scale the PSC technology and be able to achieve a feasible product in the years to come, it is mandatory to proceed with the elimination of the metal electrode. A number of factors impose this change of direction. The deposition of Au or Ag in a large-area device is both complex and costly, especially considering the deposition process and the material quantity that has to be used, and the stability of devices employing metal electrodes is under question [50–52]. Carbon materials' potential to replace metal electrodes originally stems from the work function of carbon (C) of 5.0 eV, which is close to that of gold (Au) of 5.1 eV. They are low-cost starting materials and can be used to prepare C electrodes with simple fabrication methods, and they can also contribute to the "circular" use of materials and have an additional role in the lowering of the environmental footprint, considering that they are mainly by-products of the petroleum industry. An example of this is the case of carbon black, an amorphous nanopowder obtained by incomplete combustion of carbonaceous materials, which is most widely used for the preparation of C pastes and, consequently, C electrodes [53]. In addition, C electrodes are distinguished for their high electrochemical stability, their specific hydrophobicity which contributes to the stability enhancement of the resulting devices, their inertness towards ion migration and corrosion, and their hole-transporting properties, allowing them to be incorporated into HTL-free devices. All the above facilitate the production process and lower the cost of the final devices, which is crucial when moving to large-area solar cells and modules.

### 3.2. Carbon-Based Perovskite Solar Cells (C-PSCs)

In PSCs with a carbon electrode, namely carbon-based solar cells (C-PSCs), the metal counter electrode, typically silver (Ag) or gold (Au), which is deposited by thermal evaporation, is replaced by a low-cost carbon electrode, typically prepared by a carbon paste that is coated or printed on the PSC. The feature that makes C electrodes highly promising for the upscaling of PSCs is that they can be prepared by industry applicable methods, such as screen printing, doctor blading and painting. The use of C electrodes in PSCs was first introduced by Ku et al. in 2013, when a PCE of 6.64% was obtained with a carbon black/spheroidal graphite counter electrode employed in mesoscopic heterojunction PSCs [54]. Since then, there has been considerable progress in this field and currently, C-PSCs have reached PCEs as high as 19.2%, a value which is comparable to the high-performing metal-electrode-based PSCs (Figure 9).

The differentiation of these devices originates from the processing temperature of the C starting material during the preparation of the C electrode. Thus, there are two device architectures: high temperature (i.e., HT (HT-CPSCs)) or Type I, and low temperature (i.e., LT (LT-CPSCs)) or Type II (Figure 9). HT-CPSCs comprise three prominent mesoporous oxide layers, with $TiO_2$ typically serving as the ETL, $ZrO_2$ as the spacing, an insulating layer preventing shunts from the direct contact of ETL with the C electrode, and a mesoporous C layer. The C layer is prepared from a C paste, typically consisting of graphite and carbon black along with binders in a mixture of organic solvents, which are removed after the high-temperature annealing, which also offers to the resulting C electrode the required conductivity in order to be effective. The perovskite precursor solution is then infiltrated through the triple mesoscopic stack and the perovskite film is formed after annealing. The advantages of such devices, which are also referred to as "monolithic", include their ability to be efficient without an HTL (HTL-free), that they are highly stable and that they can be prepared under ambient conditions. Moreover, they can also be entirely prepared by printing methods, which gives a head start when moving to upscaling.

LT-CPSCs' general structure is that of a standard, metal-electrode-based PSC, where there is a layer-by-layer deposition of all necessary components. An ETL, which can be either mesoporous or planar, is deposited on a compact layer/FTO substrate, followed by deposition and crystallization through annealing of the perovskite. The C electrode is deposited on the perovskite from a C paste by a simple coating technique followed by

annealing at temperatures < 200 °C. The need for a spacer insulating layer (ZrO$_2$ or Al$_2$O$_3$) is eliminated in this structure, because the perovskite layer intervenes between the C electrode and the ETL, thus preventing a short circuit at this interface. On the other hand, the composition and particularly the solvents involved in the C paste that will be used are crucial in this type of structure in order to avoid the perovskite film's decomposition. The advantage of this structure is that since the perovskite is grown directly on the ETL, there is much higher flexibility and a variety of choices considering the perovskite structures that can be utilized, while free-standing electrodes can also be implemented in the C-PSCs. This has led LT C-PSCs to exhibit the highest PCEs among PSCs with a C electrode (Table 2).

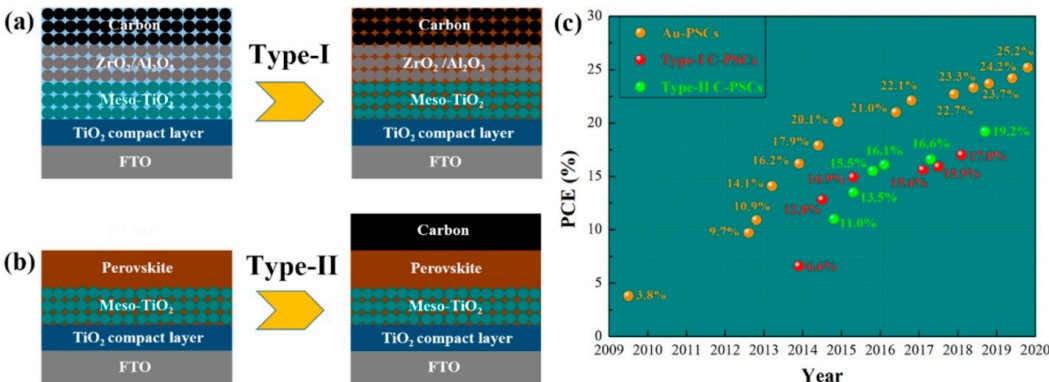

**Figure 9.** Schematics of C–PSCs configurations and efficiency evolution. (**a**) Type I C–PSCs based on a typical multi-layer mesoporous structure. (**b**) Type II C–PSCs with a structure similar to Au–PSCs. (**c**) Reported efficiency evolution of different PSCs [55]. (Reproduced from Ref. [55], 10.1016/j.carbon.2020.06.065, with permission.)

Both C-PSC device structures present advantages and drawbacks, which have been presented in detail by Bogachuk et al. in a comparative study of HT and LT-CPSCs [56].

Given that the perovskite is inserted in the HT-CPSC structure by infiltration through three mesoporous layers, the resulting perovskite crystals' size are constrained to a size <100 nm (usually 10–20 nm). Due to numerous grain boundaries, a high level of non-radiative losses occurs in this device structure, which is reflected in the limited Voc values obtained, typically <1 V, which also limit the overall performance of the device. Therefore, in order to improve the performance of HT-CPSCs, an optimized pore size is needed to provide larger perovskite crystals, and further treatments at the grain boundaries could contribute to the limiting of non-radiative recombination losses.

In LT-CPSCS, on the other hand, the layer-by-layer deposition allows for the growth of large perovskite crystals (>100 nm), by applying methods and treatments established in the highest performing metal-electrode-based devices. At the same time, Voc values exceeding 1 V have been recorded, owing to the long carrier lifetimes and high quasi-Fermi level splitting values. However, their performance is limited by low photon absorption and poor charge transport, which could be tackled by employing suitable, high-mobility HTLs and by optimizing the individual contacts.

**Table 2.** Summary of the highest PCEs obtained so far in lab-scale PSCs employing a C electrode.

| C Electrode Type | C Electrode Deposition Method | Perovskite | Hole Transport Material (HTM) | PCE | PCE Stability | Year | Ref |
|---|---|---|---|---|---|---|---|
| C film prepared from LT C paste | Press transfer | $(FA_{1-x}MA_x)PbI_{3-x}Br_x$ | Spiro-OMeTAD | 20.04 | 94% after 1000 h ambient air storage | 2022 | [57] |
| Self-adhesive C film prepared from LT C paste | Press transfer | $Cs_{0.05}(FA_{1-x}MA_x)_{0.95}PbI_{3-x}Br_x$ | Spiro-OMeTAD | 19.2 | 95% retained after 1000 h ambient air storage under atmosphere/94% retained after 80 h under 1 sun equivalent white-light-emitting diode (LED) illumination and maximum power point (MPP) in $N_2$ atmosphere | 2018 | [58] |
| C layer sprayed onto FTO substrate | Press transfer | $Cs_{0.05}(FA_{1-x}MA_x)_{0.95}PbI_{3-x}Br_x$ | Spiro-OMeTAD | 18.65 | 90% retained after 1000 h at 85°, in the dark | 2019 | [59] |
| LT paste on graphite paper | Press transfer | $Cs_{0.05}(MA_{0.15}FA_{0.85})_{0.95}PbI_{2.55}Br_{0.45}$ | Spiro-OMeTAD | 18.56 | 90.5% retained after 1500 h ambient air storage | 2020 | [60] |
| LT paste | Blade coating | 2D Octylammonium Iodide on top of 3D $FAPbI_3$ | - | 18.5 | 82% retained after 500 h of 1 sun illumination | 2022 | [61] |
| LT paste | Doctor blading | $FA_{0.3}MA_{0.7}PbI_3$ | P3HT | 18.22 | 1680 h ambient air storage/retention of ~89% after 600 h under 1 sun illumination in $N_2$ environment | 2019 | [62] |
| LT paste | Doctor blading | $(FA_{0.83}MA_{0.17})PbI_{2.15}Br_{0.85}$ | CuSCN | 18.1 | ≈95% after >2000 h under 1 sun illumination | 2019 | [63] |
| LT paste | Doctor blading | $GA_xMA_{1-x}PbI_3$ | P3HT + Ta-$Wo_x$ | 18.1 | 100% after 5000 h in a humid atmosphere | 2021 | [64] |
| LT paste | Doctor blading | $Cs_{0.05}(MA_{0.17}FA_{0.83})_{0.95}Pb(I_{0.83}Br_{0.17})_3$ | CuPc | 17.78 | ~100% retained 1200 h ambient air storage | 2019 | [65] |
| LT paste | Doctor blading | $FAPbI_3$ | $Cu_2ZnSnS_4$ | 17.71 | 30 days storage | 2020 | [66] |
| LT paste | Doctor blading | $Cs_{0.05}(FA_{0.83}MA_{0.17})_{0.95}Pb(I_{0.83}Br_{0.17})_3$ | CuSCN | 17.58 | 80% after 1000 h 1 sun illumination | 2019 | [67] |
| HT paste | Screen printing | $Cs_{0.04}(FA_{1-x}MA_x)_{0.96}PbI_{3-x}Cl_x$ | - | 17.47 | 95% of the initial PCE after 1000 h ambient air storage | 2021 | [68] |
| LT paste | Doctor blading | $Cs_{0.05}(MA_{0.17}FA_{0.83})_{0.95}Pb(I_{0.83}Br_{0.17})_3$ | CuPc | 17.46 | 97% retained after 1200 h in ambient air storage | 2018 | [69] |
| HT paste | Doctor blading | $Cs_{0.05}(MA_{0.6}FA_{0.4})_{0.95}PbI_{2.8}Br_{0.2}$ | NiO | 17.02 | 91% retained after 1000 h at 85°, in the dark | 2017 | [70] |
| HT paste | Screen printing | $(5\text{-}AVA)_xMA_{1-x}PbI_3$ | - | 16.51 | 91.7% after 1000 h continuous operation at the maximum power point under 1 sun illumination | 2020 | [71] |
| HT paste | Screen printing | $Cs_{0.1}Rb_{0.05}FA_{0.85}PbI_3$ | - | 16.26 | 360 h ambient air storage | 2020 | [72] |

| | | | | | | | |
|---|---|---|---|---|---|---|---|
| LT paste | Doctor blading | $FA_xMA_{1-x}PbI_yBr_{3-y}$ (MWCNTs added) | - | 16.25 | 93% under ambient air conditions for 22 weeks/92.7% retained after 200 h heating | 2019 | [73] |
| LT paste | Doctor blading | $CH_3NH_3PbI_3$ | P3HT | 16.05 | 90% after 1200 h upon ambient air exposure | 2021 | [74] |
| HT paste | Screen printing | $CH_3NH_3PbI_3(SrCl_2)_x$ | - | 15.9 | ≈90% after 1000 h at 1 sun illumination | 2017 | [75] |
| HT paste | Screen printing | $CH_3NH_3PbI_3$ | - | 15.89 | ~100% retained after 35 h ambient storage | 2019 | [76] |
| HT paste | Screen printing | $(5\text{-}AVA)_xMA_{1-x}PbI_3$ | - | 15.77 | N/A | 2019 | [77] |
| HT paste | Screen printing | $(5\text{-}AVA)_xMA_{1-x}PbI_3$ | - | 15.7 | 500 h under illumination, 2000 h in the dark | 2018 | [78] |

Note: Spiro-OMeTAD = 2,2′,7,7′-Tetrakis[N,N-di-(4-methoxyphenyl)amino]-9,9′-spirobifluorene, P3HT= poly(3-hexylthiophene-2,5-diyl), Ta-Wox = tantalum-doped tungsten oxide, CuSCN = copper thiocyanate, CuPc = copper phthalocyanine, NiO = nickel oxide.

## 4. Transfer to the Large Area

Even though PSCs have achieved a certified PCE of 25.7% [4] on lab-scale devices, the corresponding PCE when moving to large-area devices is far behind. In particular, according to National Renewable Energy Laboratory (NREL), the first perovskite solar module was reported by Toshiba in 2018, with a PCE of 11.7% and an active area of 703 cm², and a PCE of 11.6% in an active area of 802 cm², which are classified as "submodules". During the same year, Microquanta Semiconductor reported a mini-module with an active area of 277 cm² and PCE of 17.25%. Following this, Panasonic reported PCE of 16.1% of 802 cm² mini-module and later, in 2020, refreshed the certified PCE of 17.9% on a PSM with 804 cm² area, which is the highest PCE reported so far for PSMs (Figure 10).

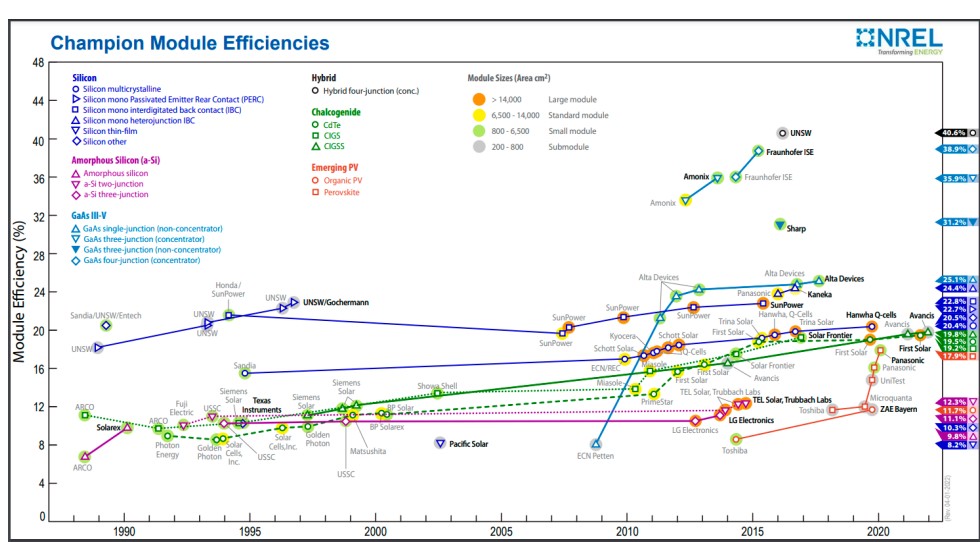

**Figure 10.** Champion photovoltaic module efficiency chart [4].

To date, lab-scale C-PSCs with active areas of ~0.05–1.0 cm² have achieved PCEs close to 20%, comparable with metal-electrode-based PSCs and with other thin-film photovoltaic (PV) technologies. Carbon electrodes present some favorable properties, such as the ability to be fabricated using a variety of carbon allotropes, which are low-cost and abundant, and their environmental friendliness and hydrophobic nature, which can make them act as a protective layer of the perovskite film towards ambient air. The combination of

the above gives great advantages to C-PSCs, including the ability to be prepared under ambient conditions, with a variety of low-cost and simple manufacturing processes and the enhanced stability of the resulting devices, which are determining considerations for modules' future applications. However, in order to be classified as a "mini-module", a PSM should have an active area of a minimum 10 cm². In particular, according to NREL, for industrial applications, solar modules are categorized as follows:

- >14,000 cm²: large module
- 6500–14,000 cm²: standard module
- 800–6500 cm²: small module
- 200–800 cm²: mini-module

The deviation between lab-scale PSCs and module-sized PSMs efficiency reports mainly include (i) the difficulty to deposit uniform charge transport and perovskite layers on large substrates, (ii) the increasing series resistance (Rs) and decreasing parallel resistance (Rp) with increasing substrate, (iii) the dead area of the sub-cell interconnections and (iv) the local perovskite degradation at the interconnection of individual cells [79]. Therefore, the determining factors in order to progress from a lab-scale device to a large-area device with minimum losses are the successful layer deposition and the appropriate interconnection of subcells.

### 4.1. Large-area C-PSC Fabrication Technologies

In general terms, the fabrication methods for perovskite solar modules are divided into two main categories, solution processing and vapor deposition techniques, that derive from the perovskite film preparation methods and are further subdivided into the following subcategories (Figure 11) [80]:

- Solution processing
- Spin coating
- Blade coating
- Slot-die coating
- Spray coating
- Inkjet printing
- Screen printing
- Vapor deposition
- Vacuum thermal evaporation
- Chemical vapor deposition
- Flash evaporation

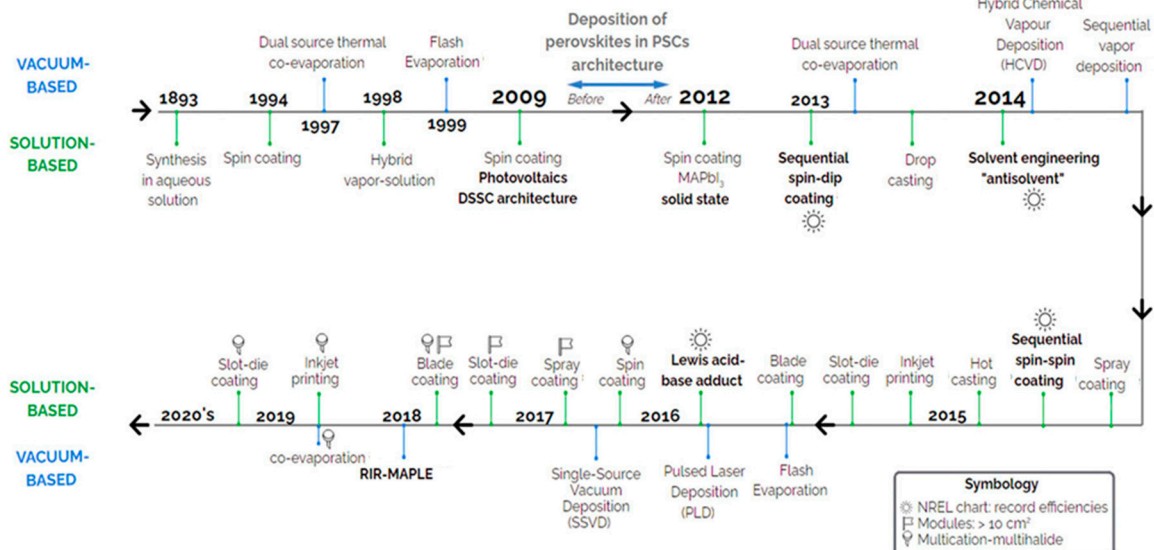

**Figure 11.** Timeline of the development of thin-film fabrication methods for inorganic and hybrid halide perovskites. The methods are classified into vacuum-based vs. solution-based. The symbols described in the inset indicate which methods were used for the fabrication of modules, multi-cation–multihalide compositions and reported NREL record efficiencies [80]. (Reproduced from Ref. [80], 10.1063/5.0027573, with permission.)

In PSMs with a C electrode, on the other hand, the classification is made with respect to the method that is used for the fabrication of the C electrode (Figure 12).

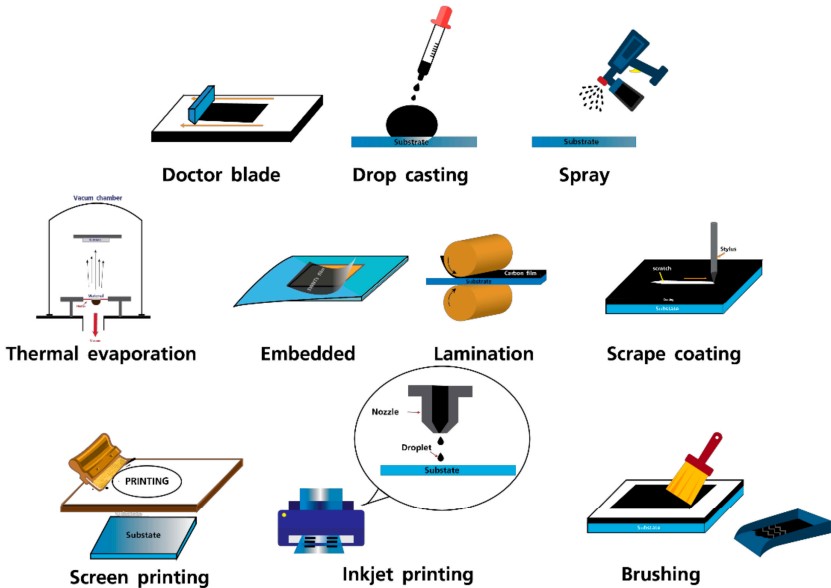

**Figure 12.** Schematic illustration of several techniques for fabrications of carbon electrodes in perovskite solar cells [81]. (Reproduced from Ref. [81], 10.3390/ma14205989, under the terms of the CC BY [4.0] license.)

The methods that have been employed, studied and reported so far are solution-based and mainly fall into the wider printing category, which includes screen printing and inkjet printing.

(a) Screen printing

Screen printing is a coating method where an ink is transferred onto a substrate in specific patterns, with the use of a mesh having specific areas made impermeable to the ink by a blocking stencil. With the use of a blade, a stencil is created on the mesh screen, pushing the ink to imprint the pattern on the underlying surface. This technique is common in fabricating screen-printed electrodes [82], and in C-PSMs, it is typically used for the layer-by-layer deposition of the mesoporous $TiO_2$ layer, the $ZrO_2$ layer and the C layer in the HT triple mesoscopic structure PSM design.

(b) Inkjet printing

Inkjet printing is a non-contact method to deposit an ink onto a substrate that consists of ink droplets that are directed continuously or on-demand on the substrate. It has been used to produce a range of printed electronic devices, such as solar panels, sensors and transistors, and it notably demonstrates better environmental performance than spin coating, owing to the nearly 0% waste production. In terms of required energy, comparing spin coating and inkjet printing, 1.8 MJ/m$^2$ and 1.152 MJ/m$^2$ have been reported, respectively, demonstrating a 36% reduction in energy consumption for inkjet printing [83,84]. In C-PSMs, it can be used to deposit the electron transport layer, the perovskite ink and the hole transport layer in all device architectures. The C electrode can also be produced by this method for C-PSMs with structures that employ an LT C paste [85].

(c) Doctor blading

The doctor blading technique, otherwise known as blade coating, is a coating method where a moving blade at a fixed yet adjustable distance from the substrate surface forms a liquid thin film by evenly spreading a pre-dispensed ink. It is a low-cost and simple method to produce films with well-defined thickness and the most widely used synthesis method for fabricating large-area perovskite films. In C-PSMs, it can be applied for the deposition of all layers (ETL, perovskite layer, HTL, C electrode) in both LT and HT device architectures.

(d) Slot-die coating

In the slot-die coating process, the precursor ink is metered through a micro fluidic metal die machine, which is positioned close to a moving substrate. It is a highly efficient deposition method in terms of material exploitation, since the waste levels are low compared to other techniques, and it is also rapid and roll-to-roll compatible for the fabrication of flexible devices. This method has been widely applied for the fabrication of organic solar cells. In C-PSMs, slot-die coating can be utilized to fabricate the ETL, perovskite and HTL layers, with the potential of also depositing the C electrode after careful regulation and optimization of C ink.

The spin coating method is also being employed for perovskite deposition; however, this method is far from ideal, considering the practical issues that arise when it needs to be applied in large substrates. Moreover, this method involves a high waste of materials (up to 95%) and the use of a high quantity of hazardous solvents; therefore, it cannot be considered as a large-area-friendly nor financially feasible deposition method. The vapor deposition processes, even though they are able to produce perovskite films that are highly uniform, require specific vacuum pressures and special equipment, increasing the operating costs. The highest PCEs that have been obtained so far in lab-scale C-PSCs were achieved by the press transfer method [57–60]; however, this method has not yet been applied in devices >1 cm$^2$.

### 4.2. Interconnection Methods

As in typical PSMs, in order to fabricate C-PSMs of a large area, the individual cells need to be interconnected, which can be either serial or parallel (Figure 13). In serial connection, a high voltage is achieved with a moderate current, while in the parallel connection, high current output is obtained with moderate voltage. Among the two, the serial connection has proven to deliver modules with higher performance [86]; thus, it is the one mainly preferred to obtain high-power-output devices.

In serial interconnection, three patterning lines (scribes), namely P1, P2 and P3, are applied on different layers of the device to make the interconnects, each one having a different role. P1 isolates the front transparent conductive oxide (TCO) of successive segments, P2 enables contact between the counter electrode and the front TCO and P3 separates the two successive segments and isolates the back contact [87].

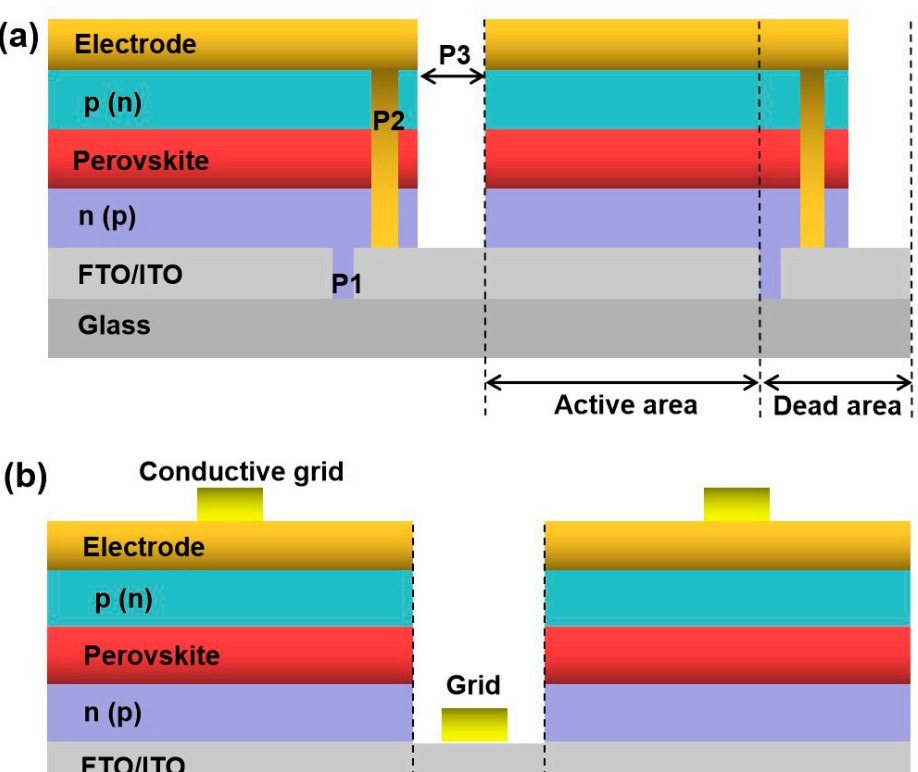

**Figure 13.** Cross-sectional schematic diagrams of grid lines. The sub-cells are connected in series (**a**) or parallel (**b**) [79]. (Reproduced from Ref. [79], 10.26599/NRE.2022.9120024, under the terms of the CC BY [4.0] license.)

The scribes also determine the active area width (Wa), the inactive, dead area width (Wd), and the contact area width. The optimization of the scribes determines the power output of the module. So far, even though mechanical scribing is also applied, the most efficient scribing method that has been used is laser scribing, owing to the low cost and ease of application of this method. An in-depth study of the scribing effect on PSM performance has been conducted elsewhere and is beyond the scope of this review [87–90]. It is worth noting though that the geometric fill factor (gFF), which strongly affects the PCE of a PSM, is directly dependent on the active area and dead area widths, defined as gFF = $\frac{\text{Active area}}{\text{Aperture area}}$, where the aperture area is the total area of the module; therefore, the scribing is a factor that should be taken into careful consideration during the design of solar modules [91].

### 4.3. Current Status—State of the Art

Taking a close look at the literature, in general terms, perovskite solar cells with an active area <1 cm$^2$ are defined as "small cells", while the term "large cells" refers to solar cells with an active area greater or equal to 1 cm$^2$. However, even though this definition is acceptable within the perovskite research community, a solar cell of 1 cm$^2$ is not acceptable for industrial applications and cannot be considered to be "large area" in practice. Therefore, even though there have been interesting approaches that have yielded promising results in C-PSCs with an active area of 1 cm$^2$, these results will only be briefly mentioned in this review, and the focus will be on the results that have been obtained in devices with a C electrode and an active area >5 cm$^2$, considering that these approaches are

more realistic and representative of the PCE losses that take place when moving from a laboratory-scale device to a mini-module and larger-sized device.

### 4.3.1. Fully Printable HT-CPSMs

In 2016, Priyadarshi et al. [92] demonstrated one of the first monolithic perovskite solar modules, of the HT triple mesoscopic structure, having active areas of 31 and 70 cm². The fabrication of the PSMs was carried out by using a semi-automatic screen printer, while the perovskite that was used was $CH_3NH_3PbI_3$ (MAPbI$_3$) with a 5-ammonium valeric acid iodide (5-AVAI) additive, which was infiltrated through a mesoporous stack. In this PSM configuration, the quality of the mesoscopic carbon layer has a huge impact on the PSM performance. The authors performed an optimization of the C layer by using different C paste formulations, keeping graphite and carbon black as the C source and varying the binder and solvent proportions, and at the same time testing a commercially available C paste. After concluding the highest performing C paste, which was the commercially available, additional optimization of the printing and the perovskite infiltration processes led to PCEs of 10.46% and 10.74% for the PSMs with active areas of 31 and 70 cm², respectively. Moreover, the PSMs presented suitable reproducibility (standard deviation = 0.65%), which is a key factor that needs to be considered in order to move the technology to the industry level, and the PSMs also exhibited 2000 h stability under ambient conditions (Figure 14).

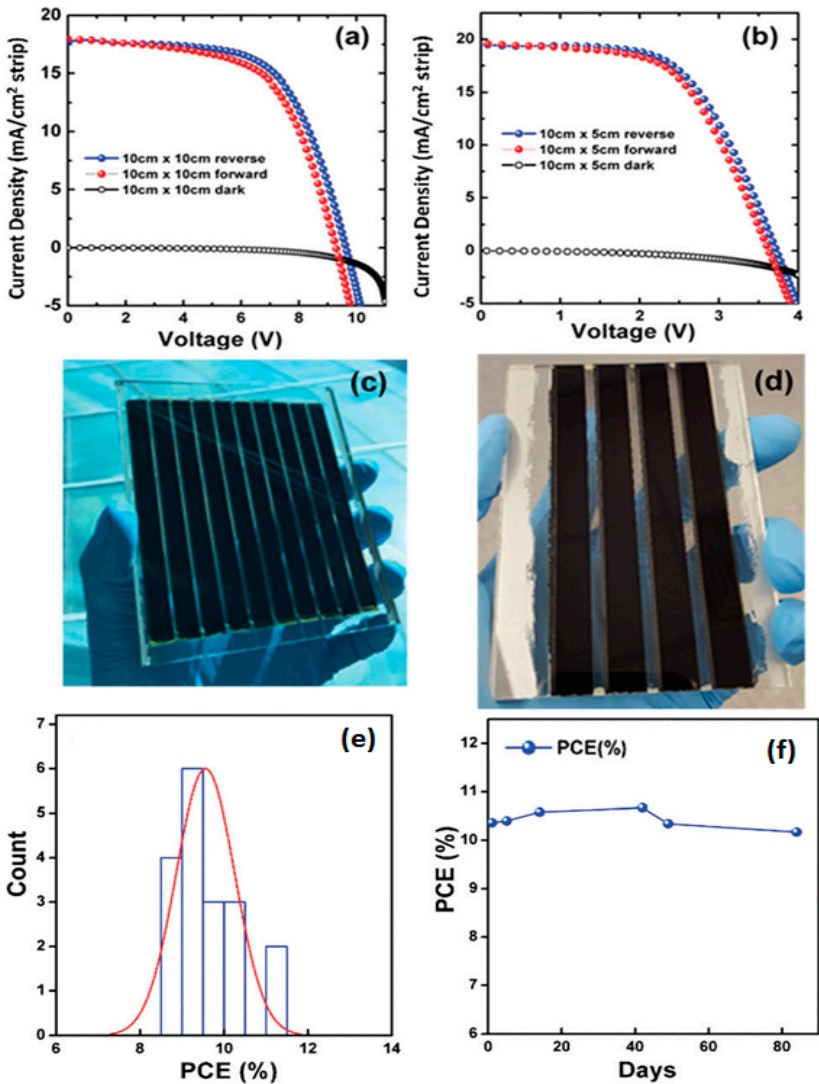

**Figure 14.** Current density vs. voltage curve under AM 1.5 G illumination: (**a**) 10 cm × 10 cm device (active area 70 cm$^2$), (**b**) 10 cm × 5 cm device (active area 31 cm$^2$), (**c**) of 10 cm × 10 cm device (**d**) 10 cm × 5 cm device. (**e**) PCE of 18 modules made in one batch and (**f**) PCE of module with time [92]. (Reproduced from Ref. [92], 10.1039/C6EE02693A, with permission.)

During the same year, the group of M. Gratzel [93] reported the deposition of the perovskite precursor in HT, HTL-free C-PSCs of the triple mesoscopic structure that were prepared by screen printing with the inkjet printing method, which they called the "inkjet infiltration of the perovskite precursor ink". This deposition method was performed using a conventional printer and was proven to be compatible with the recommended specifications of the inkjet printer from the manufacturer. This was a notable advancement in the field, considering that the manual infiltration of the perovskite in this scalable technique is a shortcoming in the upscaling process, since it causes performance variations in the fabricated devices. By mixing PbI$_2$ and CH$_3$NH$_3$I (MAI) with the 5-AVAI, which proved to effectively slow down the perovskite crystal growth, the authors managed to produce a stable perovskite precursor ink that does not clog the inkjet printer cartridge, while at the same time provides precise infiltration, thus successfully forming the absorbing layer by the printing method. Using their proposed method, which is illustrated in Figure 15, the authors prepared 10 × 10 cm$^2$ FTO–glass substrates having 18 individual cells of HTL-free triple mesoscopic C-PSCs, which yielded a mean PCE of 8.2%. Even though this cell type does not fall into the "large area" category, the proposed method is worth mentioning since it contributes to advancing the field of fully printed C-PSCs.

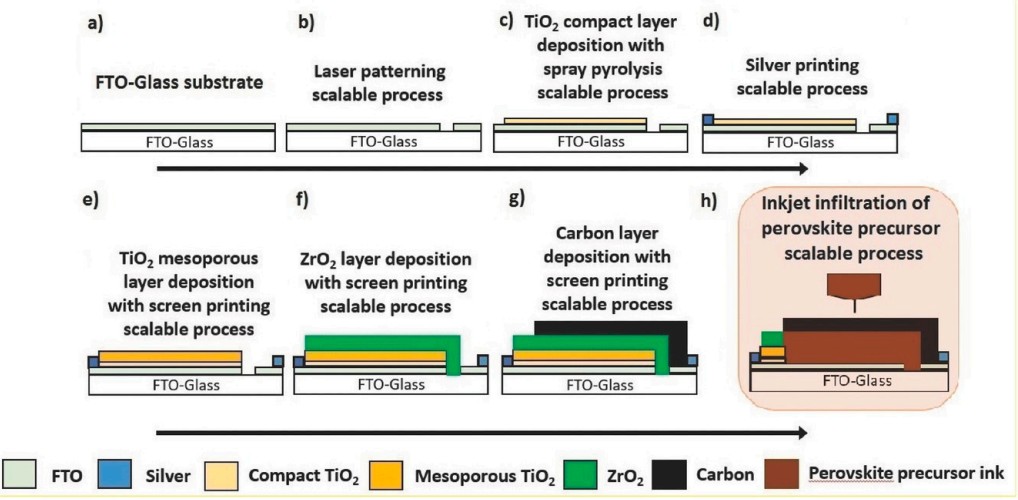

**Figure 15.** Illustration of the scalable process to be employed for the production of HTM-free carbon-based PSCs. (**a**) FTO-coated glass substrate. (**b**) Laser patterning of the FTO layer. (**c**) Spray pyrolysis for compact TiO$_2$ layer deposition. (**d**) Screen printing of silver contacts. (**e**) Screen printing of a mesoporous TiO$_2$ layer. (**f**) Screen printing of an insulating ZrO$_2$ layer. (**g**) Screen printing of a porous carbon composite layer. (**h**) Inkjet of a controlled volume of perovskite precursor solution, which is targeted in this work [93]. (Reproduced from Ref. [93], 10.1002/admt.201600183, with permission.)

A significant report that triggered interest in investigating the C electrode potential in large-scale PSCs was published in 2017, when Grancini et al. [94] demonstrated a C-PSC module with an active area of 47.6 cm$^2$ that yielded PCE of 11.2% and stability of >10,000 h with zero loss of performance, measured under controlled standard conditions and in the presence of oxygen and moisture (Figure 16). This was achieved by the addition of a small amount (3%) of 5-AVAI in the perovskite precursor, which induced the formation of a mixed two-dimensional/three-dimensional (2D/3D) (HOOC(CH$_2$)$_4$NH$_3$)$_2$PbI$_4$/CH$_3$NH$_3$PbI$_3$ ultra-stable perovskite junction composite. The modules that were fabricated are of the high-temperature (HT), triple mesoscopic

structure, which are HTL-free, low-cost and fully printable. Even though the positive effect of 5-AVAI in fully printable C-PSCs had been reported earlier by Mei et al. [95], and since then, 5-AVAI has been widely used in C-PSCs achieving high-performing PSCs, the outstanding stability of C-based modules has been highlighted in their work and has been the highest recorded value obtained for perovskite photovoltaics.

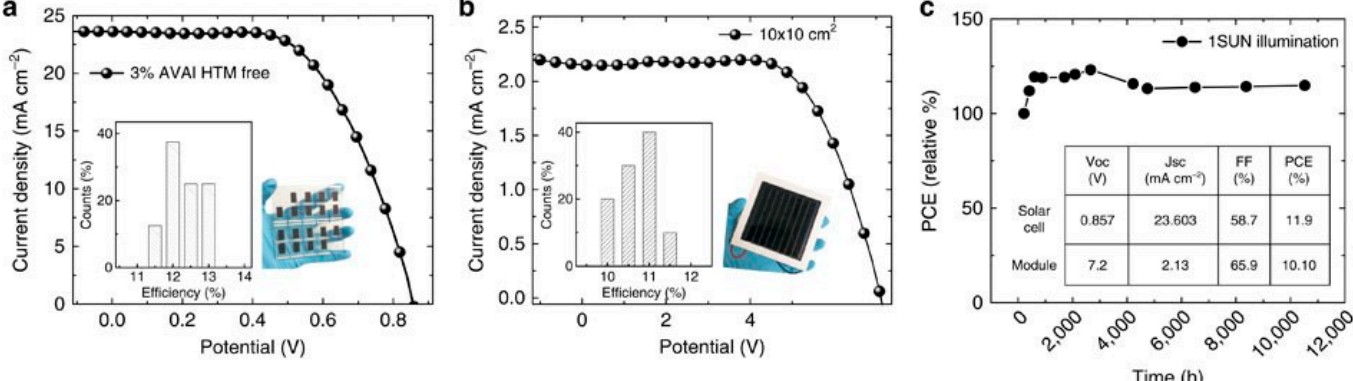

**Figure 16.** Characteristics and stability of 2D/3D carbon-based solar cells. (**a**) J–V curve using the 2D/3D perovskite with 3%AVAI in HTM-free solar cells measured under air mass (AM) 1.5 G illumination (device statistics and picture in the inset). (**b**) J–V curve using the 2D/3D perovskite with 3%AVAI in an HTM-free $10 \times 10$ cm$^2$ module (device statistics and picture in the inset). (**c**) Typical module stability test under 1 sun AM 1.5 G conditions at stabilized temperature of 55 °C and in short-circuit conditions. Stability measurements conducted according to the standard aging conditions. In the inset are device parameters of the devices represented in (**a**,**b**) [94]. (Reproduced from Ref. [94], 10.1038/ncomms15684, under the terms of the CC BY [4.0] license.)

At the same time, Hu et al. presented an HTL-free C-PSM, comprising 10 serially connected cells ($10 \times 10$ cm$^2$), of the HT triple mesoscopic structure, employing the mixed cation $(5\text{-AVA})_x(MA)_{1-x}PbI_3$ perovskite which was prepared under ambient conditions [96]. The authors explored the effect of the thickness of mesoporous TiO$_2$ and ZrO$_2$ layers on the device performance, starting with the small-area devices, where the optimal thickness of the ZrO$_2$ layer proved to be 2 μm, while for the TiO$_2$ layer, the thickness of 1 μm yielded the optimum results. By employing these optimized values, they moved their results to the large-area C-PSM, which exhibited PCE of 10.4% on an active area of 49 cm$^2$. The modules also presented adequate stability after 1000 h under illumination, after 1 month of being in a local outdoor environment and for 1 year in the dark (Figure 17).

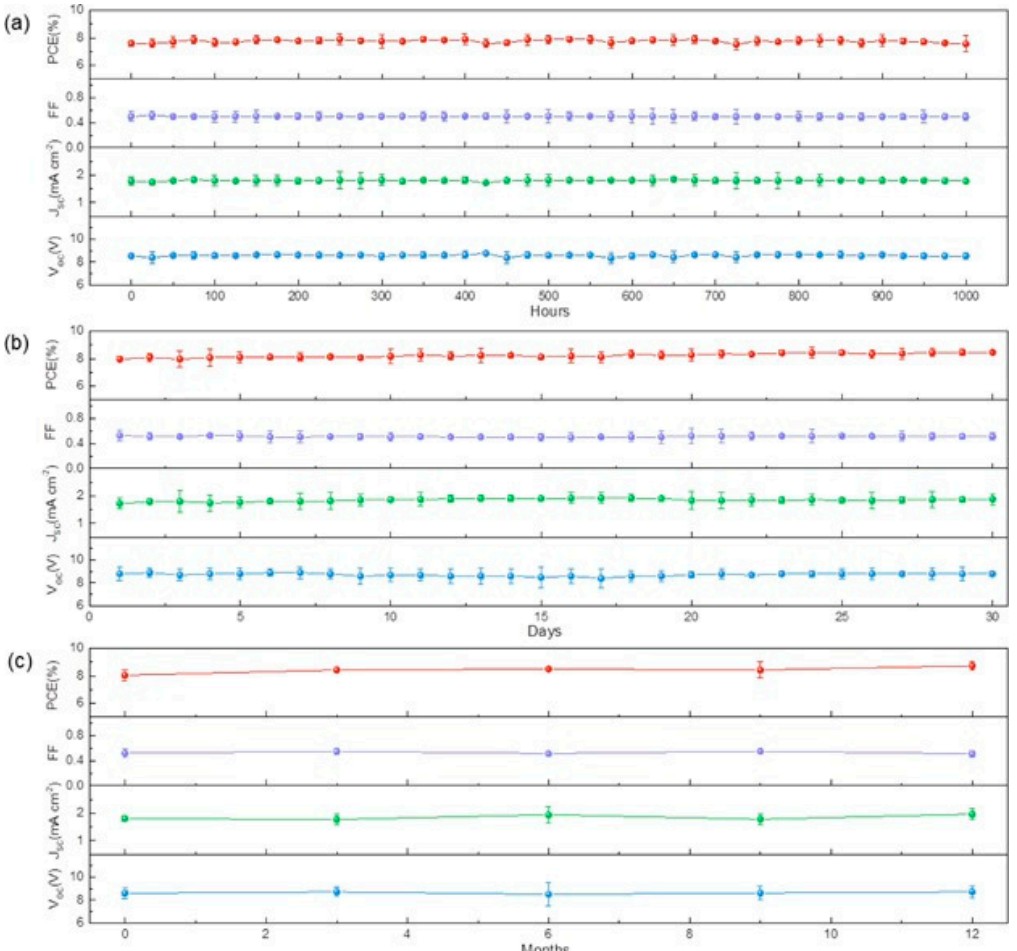

**Figure 17.** The evolution of the average photovoltaic parameters of three PSCs along with their standard deviations under different conditions: (**a**) at 100 mW/cm², (**b**) during outdoor aging in Wuhan, China, and (**c**) in the dark for 1 year [96]. (Reproduced from Ref. [96], 10.1002/solr.201600019, with permission.)

Moreover, in order to demonstrate the reproducibility of the proposed screen printing method, the authors proceeded with the fabrication of a 7 m² fully printable perovskite solar panel (Figure 18).

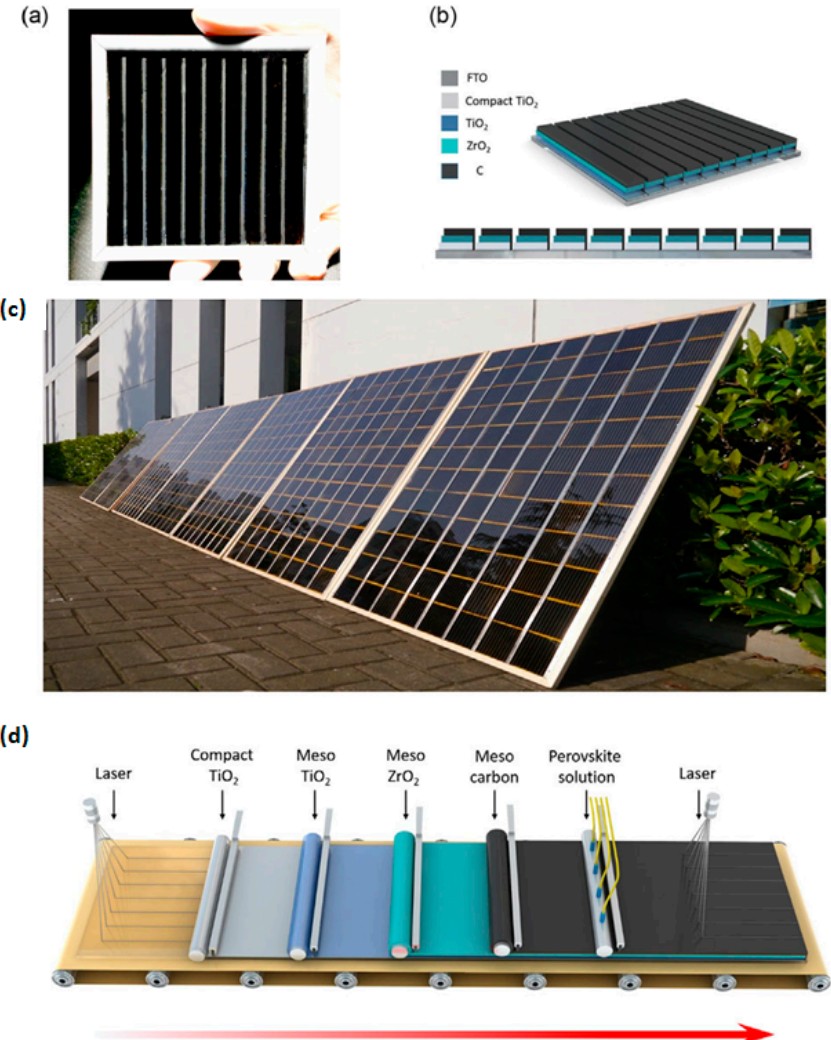

**Figure 18.** (**a**) Image of a monolithic printable PSM with 10 subcells. (**b**) The monolithic interconnection scheme of the module. (**c**) Image of 7 m² printable perovskite solar panels. (**d**) Schematic illustration of the proposed production line of PSM [96]. (Reproduced from Ref. [96], 10.1002/solr.201600019, with permission.)

An alternative approach to achieving fully printable C-PSMs was presented in 2022, when inkjet printing was proposed as an efficient method to deposit the charge transport and perovskite layers, in HTL-free C-PSMs using a mesoporous ETL and an LT C paste (LT mesoporous architecture) [97]. The authors focused on the "coffee-ring" phenomena suppression, which has so far been an obstacle towards the wide application of perovskite inkjet printing, compromising the resulting perovskite film quality and, consequently, the C-PSM performance and stability. By tailoring the $CH_3NH_3PbI_3$ perovskite precursor ink concentration in order to improve the wettability, an optimal wetting behavior was achieved and C-PSMs of 34.2 cm² active area were developed, with a mean PCE of 9.09% (Figure 19). Additionally, the C-PSMs exhibited high stability, retaining >95% of their initial PCE after 1000 h of dark storage ageing.

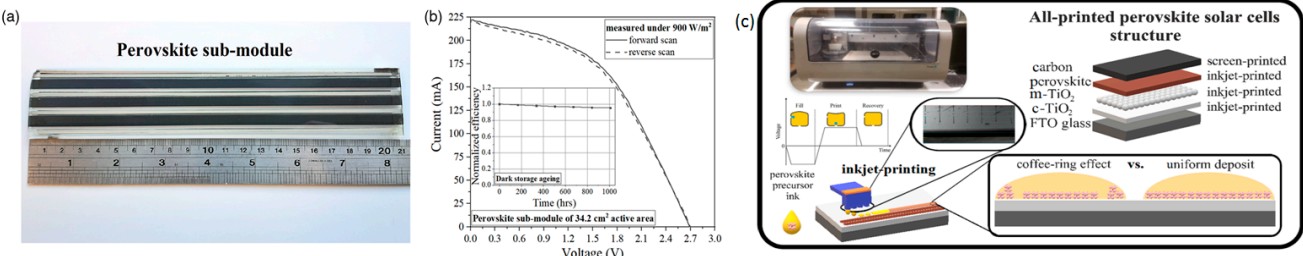

**Figure 19.** (**a**) Image of the fully printed perovskite submodule of 34.2 cm² active area. (**b**) Representative I–V curve (forward and reverse scan) of the perovskite submodule measured under 900 Wm⁻² incident solar irradiance. (**c**) Schematic illustration of inkjet printing processing of perovskite precursor inks and structure of fully printed PSCs [97]. (Reproduced from Ref. [97], 10.1002/solr.202200196, with permission.)

### 4.3.2. C Electrode Optimization

One of the reasons for the low PCE of C-PSCs compared to conventional metal-based PSCs is the poor contact of the C electrode with the underlying charge transfer films, as well as the limited charge transport ability of the C electrode. The C electrode has the ability to effectively transfer holes, and thus, C-PSCs have the advantage of being HTL-free; however, the performance of the C electrode as an HTL is lower than that of organic HTMs that are used in the highest performing PSCs, such as spiro-OMeTAD and P3HT. In order to improve the charge transport mechanism at the C–perovskite interface, Bashir et al. have introduced a p-type cobalt oxide ($Co_3O_4$) nanoparticulate thin film as an interlayer between the perovskite and the C electrode, in C-PSCs of the HT triple mesoscopic structure, prepared by screen printing [98]. With this method, the hole collection ability of the C electrode has been enhanced and the authors report ~18% improvement in the PCE of small-area PSCs, employing the (5-AVAI)CH₃NH₃PbI₃ perovskite. After transferring to the large area, they obtained PSMs with an active area of 70 cm² and PCE of 11.39% and stability of 2500 h in ambient conditions (Figure 20).

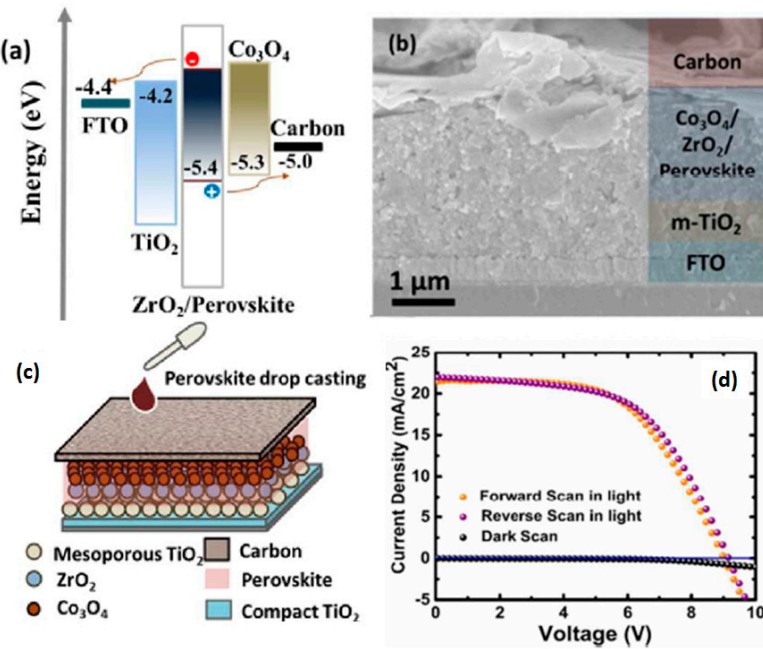

**Figure 20.** (**a**) Energy band diagram. (**b**) High-magnification cross-section SEM micrographs of the perovskite/carbon solar cell with cobalt oxide layer. (**c**) Schematic illustration of the device architecture. (**d**) J–V characteristics of standard carbon cell with $Co_3O_4$ ($Co_3O_4$/C) layer with an active area

of 70 cm² under 1 sun illumination [98]. (Reproduced from Ref. [98], 10.1039/C7NR08289D, with permission.)

The same group further increased the PCE of the PSM by altering the perovskite–C interlayer and introducing a thin layer (80 nm) of Cu-doped NiO (Cu:NiOx) nanoparticles [99]. By further improving the hole extraction efficiency and by simultaneously reducing the recombination and increasing the charge transfer efficiency, as also confirmed by electrochemical impedance spectroscopy (EIS), a PCE of 12.1% was achieved in PSMs of the HT triple mesoscopic structure, with an active area of 70 cm² employing the (5-AVAI)CH₃NH₃PbI₃ perovskite. Besides the favorable interfacial properties, the enhanced performance is also attributed to the deep lying valence band (VB) of Cu:NiOx (−5.4 Ev) forming a better ohmic contact with the perovskite layer, resulting in a large potential difference between the hole and electron transport layers and less potential losses at the HTL–perovskite interface (Figure 21). Moreover, the PSMs exhibited an increase in stability, maintaining 95% of their initial PCE value after 4500 h in ambient conditions (25 °C and 65% relative humidity (RH)) and after 60 h under continuous illumination with a constant applied bias of 0.62 V.

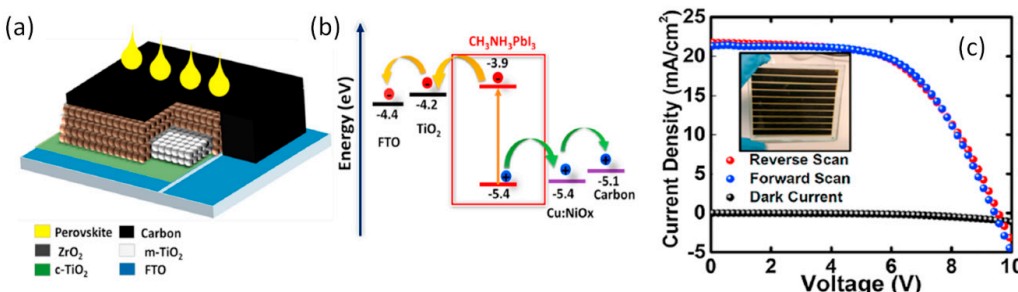

**Figure 21.** (**a**) Schematic illustration of the device architecture. (**b**) Work function of different layers. (**c**) Forward and reverse scan of device with Cu:NiOx layer with an aperture area of 70 cm² under 1 sun illumination [99]. (Reproduced from Ref. [99], 10.1016/j.solener.2019.02.056, with permission.)

One of the main causes of the efficiency gap between C-PSCs and metal-based PSCs is the low conductivity of the C electrode compared to the standard Au, Ag and Cu used in the highest performing PSCs. Raptis et al. have demonstrated a practical, simple and effective method to increase the conductivity of the C electrode by directly applying metallic grids of silver (Ag) or copper (Cu) on top of the mesoporous C layer after the perovskite infiltration, and mounting with an additional layer of low-temperature C paste, which ensures the adhesion of the grid to the underlayer and prevents delamination [100]. By employing these metallic grids in the HT triple mesoscopic structured PSMs, employing the (5-AVAI)CH₃NH₃PbI₃ perovskite, they have achieved a significant increase in the FF values for both Ag and Cu, which is the most significant limiting factor for the C-PSCs' performance and originates from the conductivity of C, and achieved PSMs of 11.7 cm² active area, with PCE values as high as 9.97% and 11.05%, respectively (Figure 22).

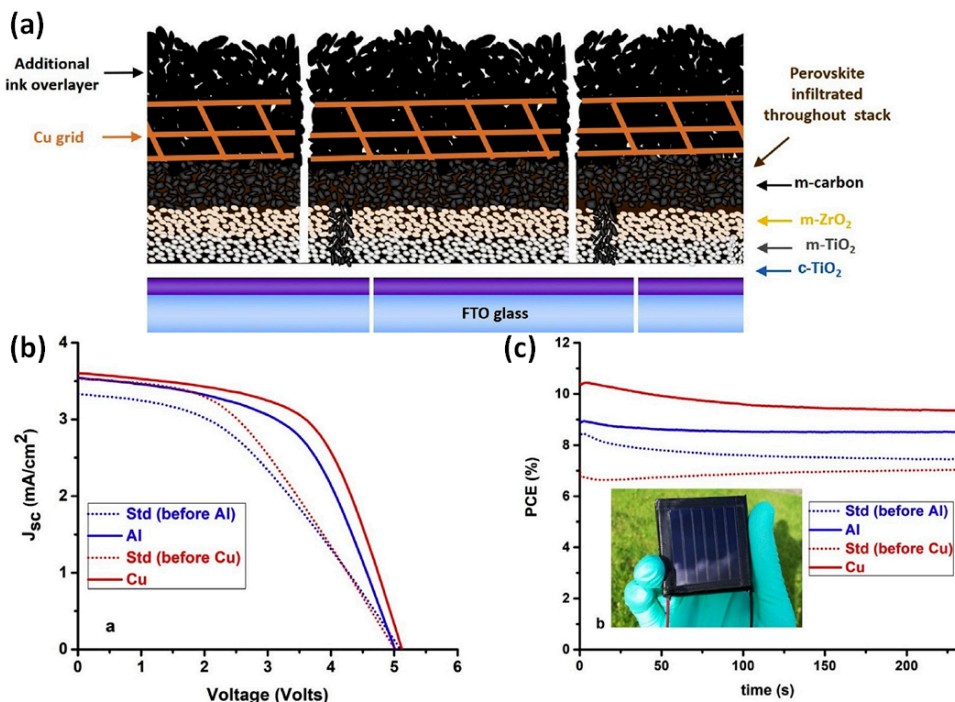

**Figure 22.** (**a**) Schematic representation of a module with copper grid. (**b**) J–V curves of the 11.7 cm²
modules before and after grid application. (**c**) Stabilized PCEs of modules before and after the placement of metallic grids. Inset shows the photo of an 11.7 cm² module [100]. (Reproduced from Ref.
[100], 10.1016/j.cap.2020.02.009, with permission.)

### 4.3.3. Device Optimization

One of the key advantages of the HT triple mesoscopic architecture regarding upscaling is its ability to be fully printable. Indeed, most of the reported C-PSMs of this architecture are fabricated by screen printing of the mesoporous layers of TiO₂, ZrO₂ and C,
under ambient conditions, achieving PCEs typically over 10% with ease and a low cost.
However, the challenge of achieving uniform deposition of the perovskite by infiltration
in large-area mesoporous scaffolds hinders the evolution of higher PCEs in this type of
device. To overcome this limiting factor, the same group used the slot-die coating method
to deposit the perovskite precursor, replacing the manual drop casting procedure [101]
(Figure 23).

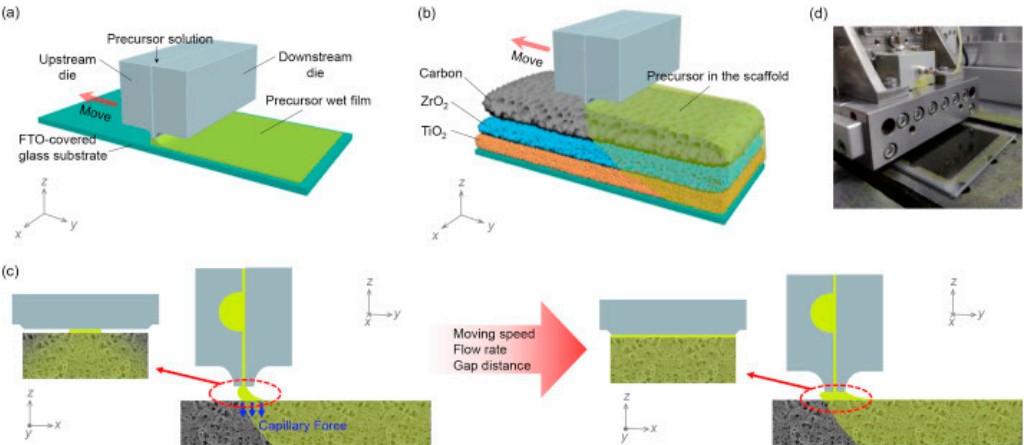

**Figure 23.** (**a**,**b**) The scheme of slot-die coating for depositing a wet film using perovskite precursors
on an FTO-covered glass substrate and on a mesoporous scaffold of TiO₂/ZrO₂/C. (**c**) The scheme of
applying the slot-die coating technique with non-optimized and optimized parameters (moving
speed, flow rate, gap distance) on a mesoporous layer. (**d**) The digital image for casting perovskite

precursor on a mesoporous scaffold of $TiO_2/ZrO_2/C$ [101]. (Reproduced from Ref. [101], 10.1016/j.nanoen.2020.104842, with permission.)

By optimizing the diffusion rate of the $(5\text{-AVAI})CH_3NH_3PbI_3$ precursor in the mesoporous scaffold, using a coating speed of 10 mm/s, the authors reported a homogeneous infiltration and a complete pore filling of the perovskite precursor in the mesoporous scaffold. The champion device exhibited a PCE of 12.87% on an active area of 60.08 cm² with a geometric fill factor (gFF) of 74.6%, which is the highest recorded up to this moment for mini-module sized C-PSMs, demonstrating slot-die coating as a promising method to fabricate highly efficient C-PSMs (Figure 24).

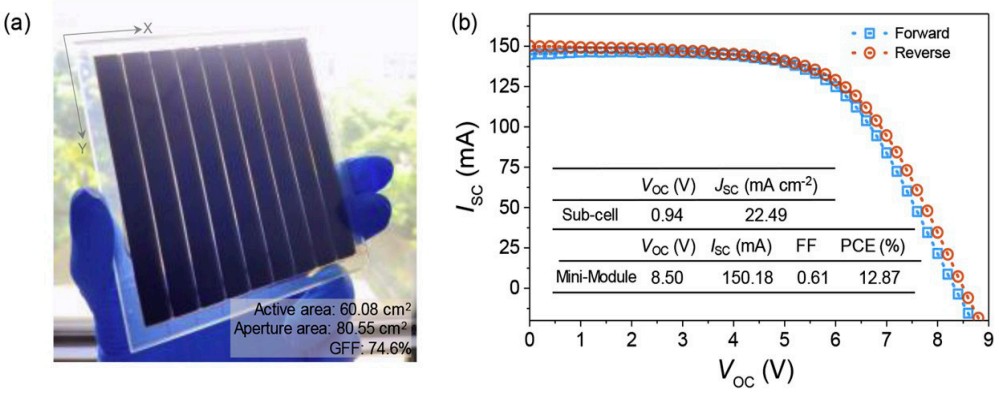

**Figure 24.** (**a**) The digital image of a triple-mesoscopic PSC mini-module. (**b**) The J–V curves of the mini-module measured at forward and reverse scans [101]. (Reproduced from Ref. [101], 10.1016/j.nanoen.2020.104842, with permission.)

One of the primary challenges of PSC upscaling is the deposition of uniform, dense, pinhole-free perovskite film, with complete coverage of the large area. In order to mitigate this issue, Cai et al. (Weihua Solar Co., Ltd., Xiamen 361101, China) applied a gas pumping method to accelerate the solvent evaporation and speed up the precipitation of perovskite, which was proven by the same group to be a facile and scalable means to deposit dense and uniform perovskite thin films [102]. The PSM consists of a ZnO compact ETL, the active perovskite layer and an LT C paste that is deposited on top of the active layer. To deposit the perovskite layer, the $CH_3NH_3PbI_3$ precursor solution was coated on the ZnO layer by slot-die coating and the substrate was placed into a sample chamber connected to the gas pump system. Using a low-pressure system (100 Pa), the rapid evaporation of the solvent in the precursor solution (dimethylformamide (DMF)) promotes the formation of the perovskite within 20 s, when the color of the film changed from yellow to brown, while the crystallization of the film was complete after annealing for 10 min. By optimizing their technique, the authors were able to achieve a PCE of 10.6% in a PSM of 17.6 cm² of active area; they also prepared modules of 45 × 65 cm² and a demonstration power station made of 32 perovskite panels (Figure 25).

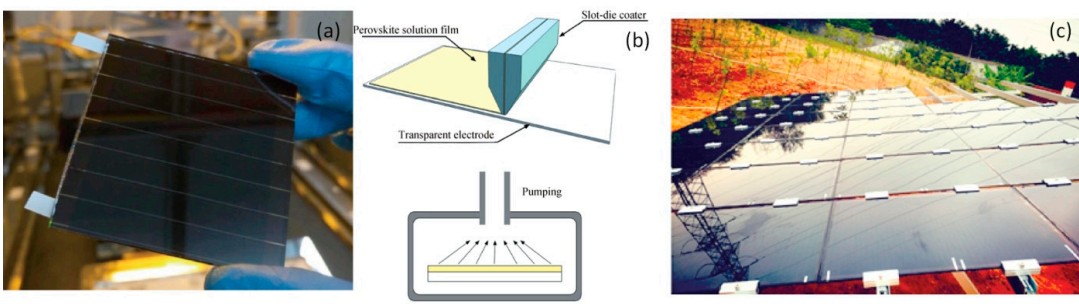

**Figure 25.** (**a**) The 5 × 5 cm² module is scribed by a laser into 8 strips, forming 7 individual cells connected in series. (**b**) Slot-die coating and pumping of perovskite liquid film and (**c**) the demonstration power plant, comprising large modules of 45 × 65 cm² [102]. (Reproduced from Ref. [102], 10.1088/1674-4926/38/1/014006, with permission.)

Besides creating dense, uniform and pinhole-free films, this process also significantly lowers the processing time, leading to less energy consumption, which would further reduce the cost of the corresponding PSM, which gives an added value to the proposed PSM fabrication method. However, a significant drop has been noted when moving from small-area (1 cm²) to large-area devices, implying that the perovskite film formation method and the device configuration still need optimizing before this process is feasible for wide application.

A significant step towards the upscaling of C-PSCs was presented by De Rossi et al. in 2018 [103], when the largest C-PSM up to that moment, of 198 cm² active area (A4 size), was demonstrated (Figure 26). The HT triple mesoscopic (TiO$_2$/ZrO$_2$/HT-C) architecture was chosen, owing to its ability to be fully printable, produced by equipment that has a low capital cost, being HTL-free and using low-cost starting materials, hence being ideal for large-scale production. The authors presented an optimization of the TiO$_2$ blocking layer (BL) in order to mitigate the practical limitations that arise from the necessity of high-temperature annealing of the sprayed BL, which causes issues in such large-area substrates, including cracking or bending, and has a direct effect on the thickness homogeneity of the printed layers. The optimization process included the turn to a low-temperature BL to ensure the integrity of the substrate on which the mesoporous layers are being printed, and at the same time, the optimization of each layer's thickness. The perovskite that was employed was the (5-AVAI)CH$_3$NH$_3$PbI$_3$, which has established stability for over 1000 h under different conditions of illumination and external stresses, and by increasing the print speed and the snap off of the screen printer, homogeneous coverage was obtained. The above adjustments combined led to a C-PSM with PCE as high as 6.6% under 1 sun illumination, while for low light intensities, the PCE rose to 11%, 17% and 18% for 200, 600 and 800 lux, respectively. Moreover, the C-PSM presented no degradation after hundreds of hours at 70% RH.

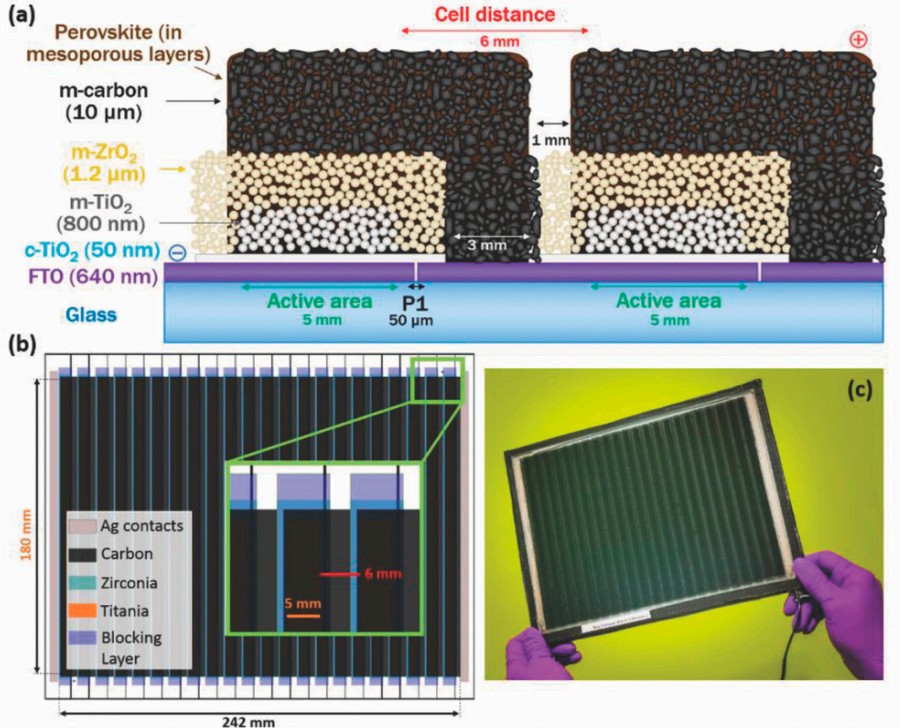

**Figure 26.** (**a**) Cross-section schematics of adjacent cells in the module with nominal thickness of each layer, highlighting the laser-etched FTO, patterning of $TiO_2$ blocking layer and the electrical vertical connection, ensured by the carbon back contact. (**b**) Module schematics, showing the different overlapping layers, the dimensions of the active area for both the individual single cell and the whole module as well as the distance between adjacent cells (inset). (**c**) Photo of a module; wires have been soldered to the silver painted busbars to provide more robust electrical contacts [103]. (Reproduced from Ref. [103], 10.1002/admt.201800156, with permission.)

Meroni et al. used the same fully printable HT triple-mesoscopic C-PSM architecture, employing the $(5\text{-AVAI})CH_3NH_3PbI_3$ perovskite, to perform an optimization of the scribing method that is used to selectively remove materials within a C-PSC [90]. Besides the material composition and tailoring and the interface engineering that govern the charge extraction and recombination mechanism and have a direct effect on a solar module's efficiency, when moving from lab-scale to large-area devices, and especially modules, another point that should be taken into account is the module design, which is also determining for the resulting PSM performance. In particular, the interconnection method can play a significant role in the geometric fill factor values. The most efficient method so far has been the scribing method, which is a material removal approach that in conventional PSMs, employing metal contacts, has achieved gFF values >90%; however, it had not been utilized for C-PSMs up until that report. After a thorough study of the PCE dependence on the dead area width (Wd), the contact area (active layer scribe—P2) and the distance between the scribes, the authors suggest that the optimal design should use a P2 width of the order of hundreds of μm and minimum safe spaces S1 and S2 (Figure 27). Their optimal design delivered C-PSMs with PCE as high as 10.29% for 5 × 5 cm$^2$ and 10.37 for 10 × 10 cm$^2$ module substrates. Their work highlights the significance of device engineering and interconnection during the upscaling of C-PSCs, as well as the necessity of scribing in order to increase the gFF of the corresponding C-PSMs in order to obtain the highest performing modules with the lowest cost.

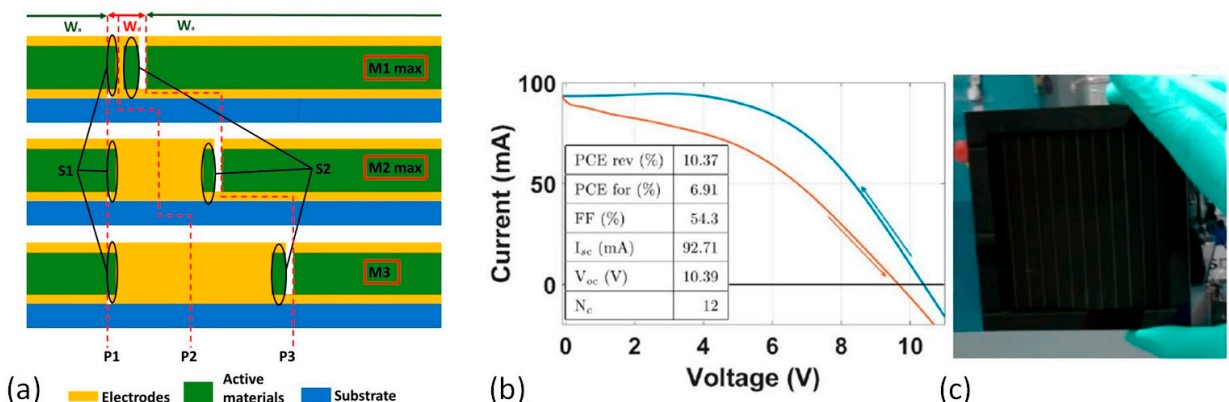

**Figure 27.** (**a**) Schematic representation of the interconnection areas for the three modules M1max, M2max and M3 with different P2 widths and minimized Wd; (**b**) current–voltage characteristic 10 × 10 cm$^2$ module with M2max design. In the inset is the table describing the device parameters. (**c**) The 10 × 10 cm$^2$ module with M2max design [90]. (Reproduced from Ref. [90], 10.3390/en13071589, under the terms of the CC BY [4.0] license.)

The same group recently presented, to the best of our knowledge, the largest C-PSM so far, with an active area of 220 cm$^2$ [104]. By adding a small amount (10%) of methanol (MeOH) in the gamma-butyrolactone (GVL)-based perovskite precursor solution, they achieved improved electrode wetting and infiltration, oriented crystal growth and higher perovskite crystal quality. The HT triple mesoscopic C-PSMs were fabricated under ambient conditions, by screen printing of the mesoporous layers and the robotic mesh method for the insertion of the precursor solution of $(5\text{-AVAI})CH_3NH_3PbI_3$ perovskite.

The scribing method was used for the module design, resulting in an active area of 220 cm$^2$ over 22 cells and a geometric fill factor of 80%, while the final PCE of the MeOH-based C-PSM increased to 9.05% compared to 8.14% of the reference C-PSM (Figure 28). Moreover, the methanol-based devices have exhibited enhanced stability, with $T_{80}$ of 4420 h at 50 °C in ambient humidity under AM1.5 illumination. Considering the large amount of solvents required for the production of solar modules, this result is of high significance, since MeOH is considered to be a more "green" solvent, while also being inexpensive and biodegradable.

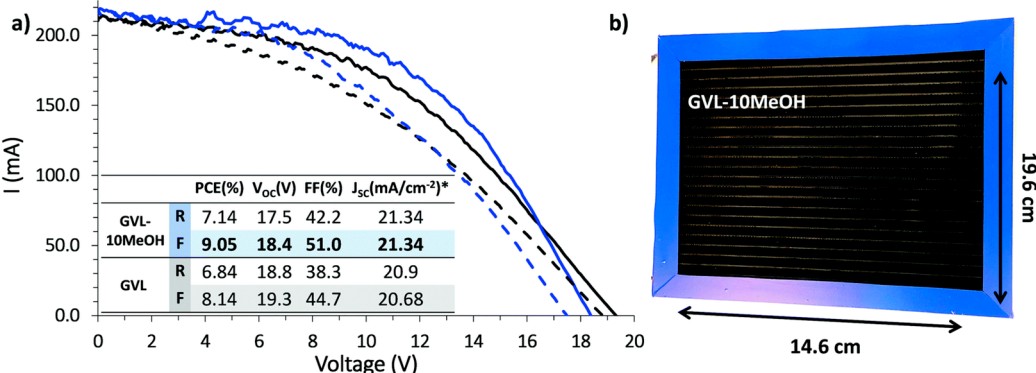

**Figure 28.** (**a**) I–V curves of modules fabricated with GVL only (black lines) and GVL–10MeOH precursors (blue lines), with inlaid table showing measured photovoltaic parameters (* Jsc as calculated for one cell of the module). (**b**) Photograph of the GVL–10MeOH module with dimensions labeled [104]. (Reproduced from Ref. [104], 10.1039/D1MA00975C, under the terms of the CC BY [3.0] license.).

It is clear so far that the most widely applied and studied structure of C-PSCs for applications in large-area devices and modules is the HT triple mesoscopic structure, owing to the ease of fabrication with a variety of methods that are scalable and compatible with industrial capabilities, combined with its low cost, originating also from the absence of HTL and the capability of ambient processing. A key component of C-PSMs of the HT triple mesoscopic structure is the mesoporous layer of TiO$_2$, which constitutes the electron transport layer. Recently, Keremane et al. [105] addressed the issue of perovskite degradation as a result of the TiO$_2$ photocatalytic activity under ultraviolet (UV) irradiation, which would potentially compromise the long-term instability of C-PSCs and PSMs and is prominent in this type of device structure. In order to circumvent the UV-activated degradation processes of the perovskite and increase both the PCE and stability of C-PSMs, the authors have proposed the modification of the active layer with Cs halides by introducing small amounts of CsI, CsCl and CsBr in the TiO$_2$ paste, expecting that Cs halides will act as active interface modifiers. Their results have shown enhanced charge extraction for all Cs-halide-modified ETLs, at the same time inhibiting of the UV-activated degradation process of the perovskite. Among the Cs halides under study, CsBr yielded devices with the highest short-circuit current density values, which are higher than the unmodified TiO$_2$, while the CsBr modification also resulted in significant improvement of the FF values, confirming the assumption of enhanced charge extraction efficiency. Finally, a high PCE of 11.55% has been recorded for a C-PSM of the triple mesoscopic structure of 70 cm$^2$ active area, fabricated by screen printing, with infiltration of the (5-AVAI)CH$_3$NH$_3$PbI$_3$ perovskite that was, however, performed in inert atmosphere. The highest PCE obtained for the reference C-PSM was 9.85%, which is ~17% lower, confirming the positive effect of the Cs halide treatment. Moreover, the stability of the C-PSMs was improved (Figure 29).

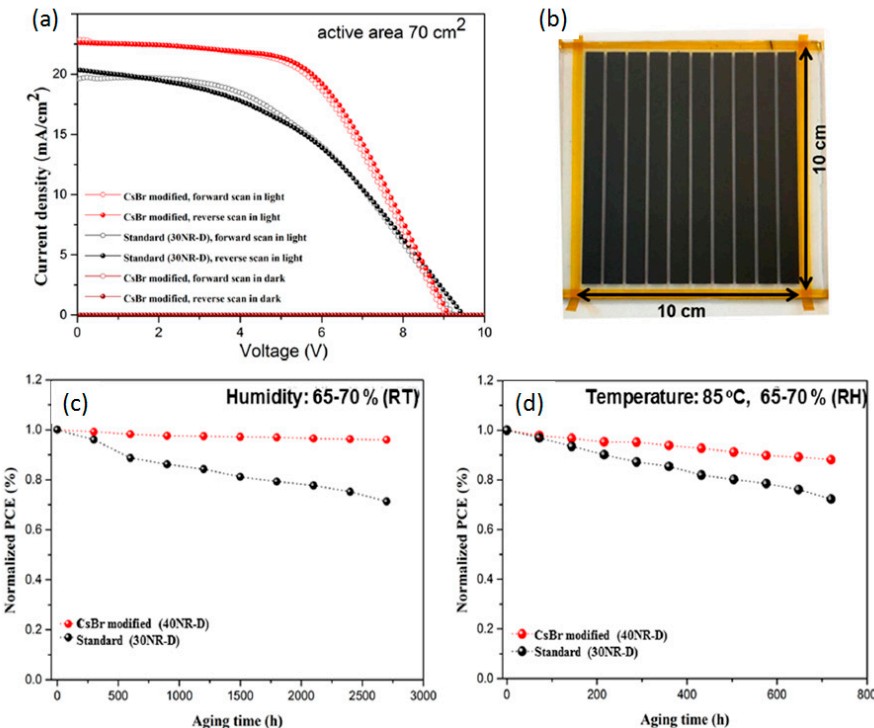

**Figure 29.** (**a**) J–V traces of unmodified and CsBr modified large-area (70 cm²) C-PSCs, recorded under 1 sun illumination and in the dark. (**b**) Photograph of the printed CsBr C-PSC module. (**c**) Normalized PCE decay as a function of time, performed under an ambient atmosphere (65–70% RH) at room temperature (RT) without sealing. (**d**) Normalized PCE decay rate at 85 °C under an ambient air for unencapsulated C-PSCs [105]. (Reprinted with permission from Ref. [105], Kavya S. Keremane, Sateesh Prathapani, Lew Jia Haur, Annalisa Bruno, Anish Priyadarshi, Airody Vasudeva Adhikari, and Subodh G. Mhaisalkar, ACS Applied Energy Materials 2021 4 (1), 249–258 DOI: 10.1021/acsaem.0c02213 Copyright© 2023 American Chemical Society.)

### 4.3.4. Stability

As previously mentioned, the most widely applied C-PSM architecture is the HT triple mesoscopic architecture, because of the multiple advantages that this architecture possesses towards commercialization. Bogachuk et al. performed a study of the stability of these printable C-PSMs under reverse bias stress which certified their remarkable durability [106]. Even though these cells have a breakdown at −3.6 V, they have proven to withstand at least 60 min under −8 V and 30 min under −9 V, stemming from the robustness of the C electrode, which does not melt at high temperatures, combined with the lack of metal ion migration from the electrode to the perovskite and the inertness of the electrode that does not oxidize. Moreover, the authors have proven that even though C-PSMs undergo iodine loss under mild voltages, this is a reversible reaction, contrary to the permanently destroying degradation. The screen-printed HT triple mesoscopic C-PSMs that were prepared had an active area of 56.8 cm², a gFF of 93.4%, employed the (5-AVAI)CH₃NH₃PbI₃ perovskite and exhibited a PCE of 11.1% (Figure 30). In addition, the C-PSMs passed the "hotspot" test described in the International Electrotechnical Commission (IEC) 61215:2016 international standard at an accredited module testing laboratory, for the first time, which is a significant step towards their industrialization.

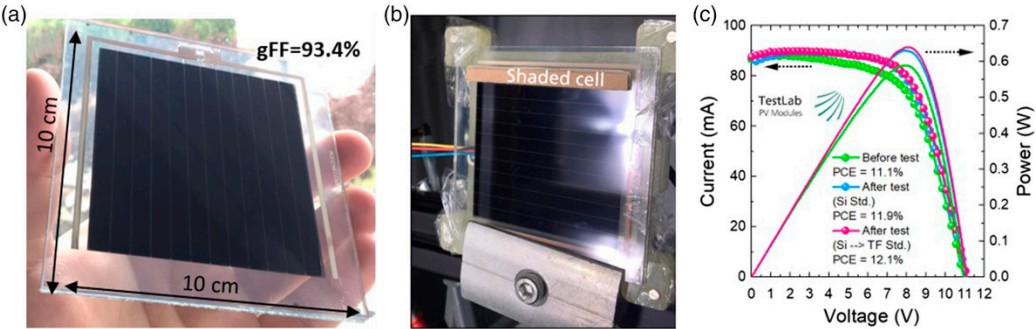

**Figure 30.** (**a**) Photograph of an encapsulated perovskite solar module with carbon-based contact (C-PSM). (**b**) Photograph of the module with one fully shaded cell during the hotspot test. (**c**) I–V and P–V curves of the module before (green) and after conducting test according to silicon ageing test (blue), followed by conducting an additional thin-film ageing test (cerise) on the same module, showing no loss of performance after both IEC-compliant tests [106]. (Reprinted with permission from Ref. [106], 10.1002/solr.202100527, under the terms of the CC BY [3.0] license Copyright © 2023Wiley-VCH GmbH.)

### 4.3.5. LT-CPSMs

One of the challenges regarding the upscaling of C-PSCs is the ability to prepare uniform, dense and pinhole-free large-area perovskite films. The same PSM architecture, employing a mesoscopic $TiO_2$ layer as the ETL, with no $ZrO_2$ and HTL employed, and with the use of an LT C paste to fabricate the counter electrode, was used in a study presented by Lou et al. that focuses on the improvement of perovskite films in a large area [107]. The antisolvent treatment method was applied, which is well known to produce perovskite films of high quality by increasing the nucleus density during the film formation [108]. By tuning the coordinated solvent number (CSN) in the intermediate phase, through solvent engineering, the perovskite nucleation and growth were regulated, producing high-quality perovskite films. A mixture of ethyl acetate (EA) and toluene (Tol) was used as an antisolvent and C-PSMs with the $CH_3NH_3PbI_3$ perovskite absorber of the LT mesoscopic architecture having an active are of 52 cm$^2$ were fabricated. The C-PSM fabrication was performed by spin coating of the active layers in inert atmosphere, while the C electrode was deposited by doctor blading. The champion C-PSM exhibited a PCE value of 10.2% and better stability compared to the reference device, retaining 95% of its initial efficiency after 500 h of storage under 40% humidity conditions (Figure 31). Even though the method reported is a simple and effective way to produce high-quality perovskite films, both techniques of antisolvent dripping and spin coating have been questioned regarding their applicability to large areas, considering the elevated costs due to the material waste of these methods, as well as the environmental impacts.

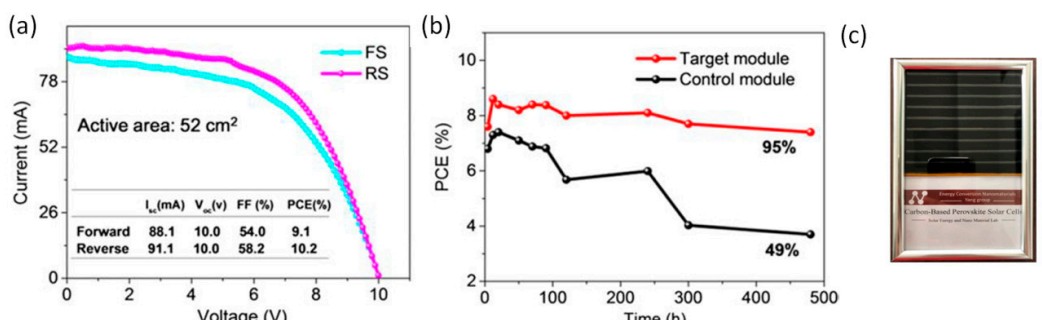

**Figure 31.** (**a**) The corresponding I–V curves of the champion PSM with forward and backward scanning under simulated standard AM1.5G sunlight. (**b**) PCE degradation curves with storage time under ambient conditions and 40% RH for different PSMs. (**c**) Photograph of a 10 cm × 10 cm PSM [107]. (Reproduced from Ref. [107], 10.1002/ente.201900972, with permission.)

He et al. presented the upscaling of a different C-PSC architecture, LT planar [109], motivated by the lower cost of planar structures—which, by not employing a mesoporous scaffold, do not require high-temperature annealing—and by their compatibility with flexible substrates. In their work, with the scope of enhancing the hole transport properties of the C electrode, to optimize the perovskite–carbon interface and at the same time to fabricate all low-temperature-processed C-PSCs, they proceeded with the modification of the C electrode by introducing in the C paste the hole-transporting copper phthalocyanine (CuPc) as an additive. CuPc has proven to act both as an interface modifier and a dopant, and the enhanced hole extraction from the perovskite to the C electrode has been attributed by the authors to an enhanced electrical field at the perovskite–modified C interface. C-PSMs with the structure of FTO/compact $TiO_2$/$CH_3NH_3PbI_3$/C and with active area of 22.4 $cm^2$ were fabricated employing the CuPc-modified C electrode and exhibited a PCE of 7.2%, while the replacement of FTO with a flexible conductive substrate yielded flexible C-PSMs with PCE of ~6% (Figure 32).

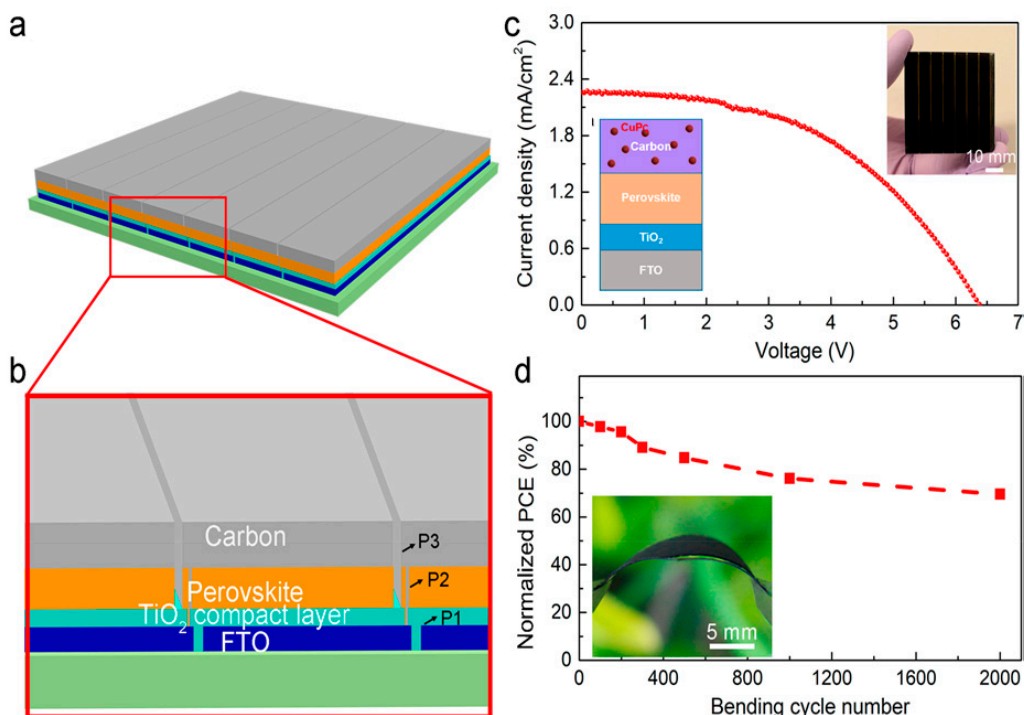

**Figure 32.** Scalable fabrication of the carbon-based PSC modules. (**a**,**b**) Schematic figure of the module structure. (**c**) J–V curve of the as-prepared PSC module on a 5 cm × 5 cm FTO substrate with 22.4 $cm^2$ designated area (inset: photograph of solar module with the carbon electrode). (**d**) Dependence of the PCE change of the PSC on a flexible substrate on the increasing bending cycle number (inset: photograph of a solar module in bending conditions) [109]. (Reproduced from Ref. [109], 10.1021/acsenergylett.9b01294, under the terms of the CC BY-NC-ND [4.0] license.)

The low-temperature planar architecture has also been employed by Yang et al. [64] to fabricate a four-cell PSM with an active area of 4 $cm^2$ (on a 25 $cm^2$ substrate), achieving a highly promising PCE of 15.3%. The PSM with the ITO/$SnO_2$/perovskite/P3HT/C configuration was fabricated with the blade coating method and was fully printed (Figure 33). The novelty of their work is based on the perovskite optimization, besides the low-temperature, scalable, fully printable protocol that was proposed. First, they optimized the perovskite precursor solution by replacing the commonly used mixed dimethyl formamide (DMF)/dimethyl sulfoxide (DMSO) solvent with a mixture of 2-methoxyethanol (2ME)/N-methyl pyrrolidone (NMP). With this alteration, they managed to increase the uniformity and surface coverage of the perovskite film. Secondly, the guanidinium

chloride (GuCl) additive was used in a small proportion (10%) in the CH₃NH₃PbI₃ perovskite, which has proven to promote the grain growth, increasing the perovskite grain size and decreasing the trap density, therefore leading to improved crystallinity and morphology of the resulting perovskite film. The combination of these parameters led to the highest PCE reported so far for a PSC of active area >1 cm².

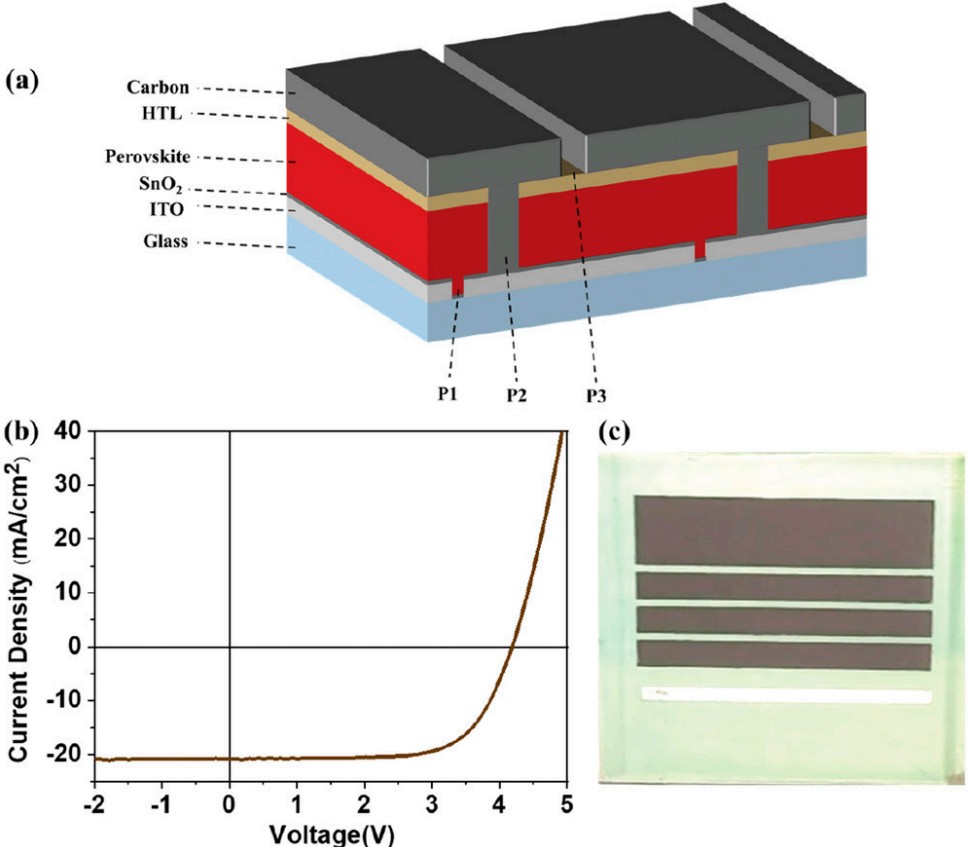

**Figure 33.** (**a**) Schematic diagram of the carbon electrode perovskite solar module. (**b**) J–V curves of the carbon PSM on the 25 cm² substrate. (**c**) The digital image of planar carbon PSM [64]. (Reproduced from Ref. [64], 10.1002/aenm.202101219, with permission.)

The above results summarize the current state of the art in perovskite solar modules with a C electrode (C-PSMs) and active area >5 cm². Even though we consider the results that have been presented as "mature" enough to potentially become a product, and therefore close to commercialization, we cannot disregard the significant work that has been accomplished in smaller "large area" PSCs (active area <5 cm²). Table 3 presents some promising approaches that have been proposed for PSCs with active areas ranging from 1 to 3.5 cm².

**Table 3.** Summary of the highest PCEs obtained so far in PSCs employing a C electrode with active area 1–3.5 cm².

| Active Area (cm²) | Fabrication Method | Structure (with Respect to C Electrode) | Perovskite | PCE (%) | Comments | Ref. |
|---|---|---|---|---|---|---|
| 1 | Spin coating—C paste printing | LT mesoporous | CH₃NH₃PbI₃ | 9.72 | IPA/CYHEX solvent in perovskite precursor solution | [110] |

| 1 | Spin coating—C paste doctor blading | LT mesoporous | $CsPbI_{2.5}Br_{0.5}$ | 13.35 | F-doped carbon quantum dots in the perovskite layer | [111] |
|---|---|---|---|---|---|---|
| 1 | Spin coating—C paste doctor blading | LT mesoporous | $CH_3NH_3PbI_{3-x}Br_x$ | 12.5 | NiO-C intermediate layer | [112] |
| 1 | Spin coating—C paste blade coating | LT planar | $Cs_{0.04}(MA_{0.17}FA_{0.83})_{0.96}Pb(I_{0.83}Br_{0.17})_3$ | 15.18 | Inverted structure/flexible device/Cr buffer layer | [113] |
| 1 | Spin coating—C paste doctor blading | LT mesoporous | $CH_3NH_3PbI_3$ | 11.08 | N, O co-doped biomass porous composite carbon electrode based on activated soybean dregs carbon (SDC) | [114] |
| 1.5 | Inkjet printing—C screen printed | HT triple mesoscopic | $CH_3NH_3PbI_3$ | 9.1 | Fully printed | [115] |
| 1 | Spin coating—C paste doctor blading | LT mesoporous | $CH_3NH_3PbI_{3-x}Cl_x$ | 13.04 | C + $CH_3NH_3I$ as HTM | [116] |
| 1 | Screen printing | HT triple mesoscopic | (5-AVAI) $CH_3NH_3PbI_3$ | 13.99 | additional hot-pressed highly conductive LT carbon layer on the back carbon electrode | [117] |
| 3.49 | Spin coating—C paste doctor blading | LT mesoporous | $Cs_{0.05}(FA_{0.85}MA_{0.15})_{0.95}Pb(I_{0.85}Br_{0.15})_3$ | 13.86 | Spiro-OMeTAD HTM/graphene-doped $TiO_2$ | [118] |
| 1 | Spin coating—C paste doctor blading | LT mesoporous | $CH_3NH_3PbI_3$ | 13 | PTAA HTM in the antisolvent | [119] |
| 1 | Spin coating—press transfer | Planar n-i-p | $Cs_{0.05}(FA_{0.85}MA_{0.15})_{0.95}Pb(I_{0.85}Br_{0.15})_3$ | 14.05 | C cloth film electrode/spiro-OMeTAD HTM/flexible device | [120] |
| 1 | Spin coating—press transfer | Planar n-i-p | $Cs_{0.05}(FA_{0.85}MA_{0.15})_{0.95}Pb(I_{0.85}Br_{0.15})_3$ | 17.02 | carbon film electrode composited with graphite paper/spiro-OMeTAD HTM | [60] |

Note: IPA = isopropanol, CYEX = cyclohexane, KOH = potassium hydroxide, PTAA = polytriarylamine.

*4.4. Cost Analysis*

One of the greatest advantages of C electrodes incorporated in PSMs compared to metal electrodes is their low cost, which is crucial for the introduction of a technology to the PV market. C-PSMs are produced with inexpensive starting materials, with low-cost processes such as printing and in ambient air conditions. A cost performance analysis has been performed by Cai et al. [121] which determined that the module costs for PSCs are one-third of the cost of bulk silicon (Si) PVs. In particular, they examined the fabrication of two types of modules: a moderate-efficiency (~15%), fully printable module, prepared by an extremely low-cost, "humble" process, with power output of 120 W, and a high-efficiency (~20%) module prepared by high-cost, "noble" processes and a power output of 190 W. The total material cost of the fully printed module, considering a module surface area of 1 m² and annual capacity of 100 MW, was calculated at 0.127 USD/W, the overhead cost as 0.098 USD/W, the amortizing cost as 0.25 USD/W and the final module cost as 0.28–0.25 USD/W. They demonstrated that the levelized cost of electricity produced (LCOE) is

3.5–4.9 US cents/kWh with a 15-year lifetime, which surpasses the U.S. "SunShot Initiative" target of 6 US cents/kWh, highlighting the potential of this technology.

Song et al. performed a specific economic evaluation of a low-cost, printable PSC by using the "bottom-up cost modeling" that is being used by several PV manufacturing companies [122]. The total manufacturing cost has been calculated as 31.7 USD/m² and an average cost of 0.41 USD/Wp has been calculated for a module with 16% efficiency, while the obtained LCOE is 5.82 US cents/kWh with a 30-year lifetime. Even though the fabrication process is similar to the one applied for C-PSMs, their modules employed Al back contacts, which differentiates the results.

A more representative work has been recently reported by Kajal et al., where the authors applied bottom-up cost modeling to perform a cost analysis of C-PSMs [123]. In a detailed study, they evaluated and compared the manufacturing cost of the two types of C-PSMs (high temperature, Module A; low temperature, Module B). The C-PSMs were designed to be 1 m² with 20 monolithically integrated cells and have a PCE that is 75% of the lab-scale cell PCE, taking into consideration the losses that occur when moving from small- to large-area devices. The average production cost has been calculated as 0.21 USD/W and 0.15 USD/W for Modules A and B, respectively, and the LCOE ranged from 0.034 to 0.016 USD/kWh for Module A and from 0.030 to 0.014 USD/kWh for Module B, with lifetime variation from 10 to 25 years while retaining 80% of the initial PCE. Their analysis has also highlighted that the cost of starting materials significantly drives the module cost. For Module A, it was observed that the $ZrO_2$ is the top contributing factor to the cost. It has been observed for Module B that it has the highest sensitivity to most of the factors that determine the cost (Figure 34).

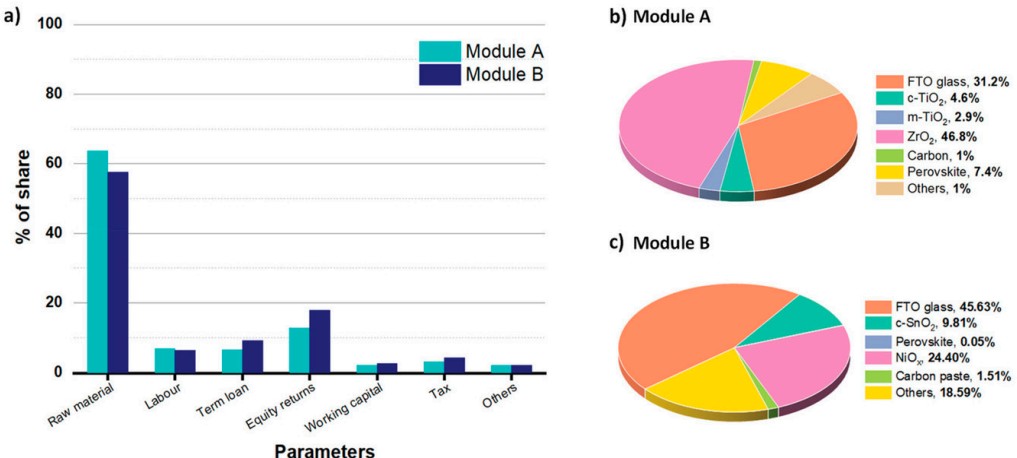

**Figure 34.** (**a**) Different parameters and their share analysis for both modules, (**b**) material share for Module A, and (**c**) material share for Module B [123]. (Reprinted with permission from Ref. [123], 10.1002/gch2.202100070, under the terms of the CC BY [3.0] license Copyright © 2023 Wiley-VCH GmbH.)

The above results and estimations are comparable to the currently widely produced and applied Si solar modules. This is significant, considering that C-PSMs emerged as promising less than a decade ago, and the intensive optimization of the materials and methods, as well as module design and architecture, is currently in progress.

## 5. Challenges and Future Prospects

It has been made clear that the application of C electrodes in PSMs can make this technology able to be transferred to the commercial stage. The acknowledgement of this potential is also evident by the increasing rate of papers that have addressed this subject over the past 5 years. The potential of C-PSMs has been presented, but important challenges remain that need to be tackled, both towards the PCE and stability enhancement

and to contribute to the feasibility of C-PSM as a product in financial terms. The most important factors that still need to be studied and improved are as follows.

(i) The homogeneity of prepared films when the C-PSC structure is transferred to a large area. On the device level, the loss in PCE when moving from a solar cell to a solar module is a significant barrier to the commercialization of C-PSMs. One of the causes is the difficulty of depositing homogeneous layers on large substrates by using industrially compatible and low-cost methods, such as printing. The main challenge regards the perovskite layer, which has the most determining effect on the device performance. Further research should be directed toward the optimization of the deposition and annealing methods that will produce high-quality films under ambient conditions. Chemical modifications as well as tailored engineering methods that have already been tested in other thin-film technologies could contribute to this end. By obtaining homogeneous and highly crystalline charge extraction and perovskite layers, the PCE of C-PSMs can be further improved and be comparable with Si-based PV modules, with the additional advantage of much lower production costs.

(ii) The interconnection method that further contributes to the performance improvement. As previously mentioned, the interconnection of the cells, which is defined through scribing, during the upscaling is directly connected to the geometric fill factor, which is a key variable during PV modules' manufacturing. Laser scribing, as opposed to mechanical scribing, has emerged as highly promising, and it can also be effective when combined with the monolithic fabrication of modules using methods such as screen printing. By benefiting from the advances in laser applications for electronics manufacturing, the scribing method can be optimized, providing high gFF values and further improving the PCE while minimizing the losses of upscaling.

(iii) The encapsulation of C-PSMs in order to protect the modules from environmental conditions and mechanical damage. Proper encapsulation of the modules, combined with C-PSMs' high stability, can achieve a product that will have a decade-long lifetime, like the PV modules that are currently dominant in the PV market. This requires intensive research, as C-PSMs are multilayered constructions, with the individual layers being sensitive to different temperature and chemical variations, with the perovskite layer being the most prone to degradation.

By resolving the above issues PSMs with a C electrode can make an entrance to the PV market, which is predicted to occur soon.

## 6. Conclusions

Carbon electrodes have long been considered as highly promising for applications in energy-harvesting devices, particularly next-generation solar cells, owing to their exceptional electrical and catalytic properties. High power conversion efficiencies have been recorded in a variety of emerging photovoltaic technologies, and in the past 5 years a notable turn to C electrodes has been made in perovskite solar cell research. Besides high-efficiency, small-area devices, C electrodes have been successfully implemented in perovskite solar modules of low cost, high stability and scalability. In this review, the background of C electrode applications in solar cells has been presented and the highest performing C-electrode-based small-area perovskite solar cells have been summarized, which now boast PCE values exceeding 19%, with exceptional stability, constituting a strong motivation for the upscaling of this type of device. The current status and state of the art of C-electrode-based PSMs have been presented and the challenges of upscaling have been discussed, along with the cost benefits of this technology.

C-PSMs, which can be fabricated with a series of low-cost, industrially available and affordable methods, such as printing, have shown their potential in the transition of this technology from the lab to the market. However, higher PCEs and stability can be further achieved by optimizing (a) the conditions of perovskite deposition, in order to obtain homogeneous active layers on large substrates; (b) the interconnection of subcells through appropriate scribing, which will maximize the geometric fill factor; and (c) the

encapsulation method, to ensure a guaranteed functionality of the solar module, comparable with the modules that currently dominate the market. Moreover, deeper cost analysis needs to be performed, which should include a variety of parameters that determine the feasibility of a product. Overall, PSMs with C electrodes appear to be the most viable type of PSCs to be introduced to the market, as also evident from the few but important and successful companies that have pursued this venture.

**Funding:** This research received no external funding.

**Conflicts of Interest:** The authors declare no conflicts of interest.

**Abbreviations**

| | |
|---|---|
| 5-AVAI | 5-Ammonium valeric acid iodide |
| Ag | Silver (element) |
| Al | Aluminium (element) |
| $Al_2O_3$ | Aluminium oxide |
| A.M. | Air Mass |
| Au | Gold (element) |
| BHJ | Bulk HeteroJunction |
| C | Carbon (element) |
| C-PSC | Carbon Perovskite Solar Cell |
| C-PSM | Carbon Perovskite Solar Module |
| CB | Carbon black |
| CdSe | Cadmium Selenide |
| CE | Counter Electrode |
| CNH | Carbon nanohorn |
| CNF | Carbon nanotube fibre |
| CNT | Carbon nanotube |
| Co | Cobalt (element) |
| CsBr | Caesium Bromide |
| CsCl | Caesium Chloride |
| CsI | Caesium Iodide |
| CTL | Charge Transport Layer |
| Cu | Copper (element) |
| CuPc | Copper phthalocyanine |
| CuSCN | Copper(I) thiocyanate |
| DMF | Dimethylformamide |
| DSSC | Dye Sensitized Solar Cell |
| EIS | Electrochemical Impedance Spectroscopy |
| ETL | Electron Transport Layer |
| FA | Formamide |
| FAI | Formamidinium Iodide |
| FF | Fill Factor |
| FTO | Fluorine-doped Tin Oxide |
| HOIP | Hybrid Organic Inorganic Perovskite |
| HT | High Temperature |
| HT-CPSC | High Temperature Carbon Perovskite Solar Cell |
| HTL | Hole Transport Layer |
| HTM | Hole Transport Material |
| In | Indium (element) |
| $J_{sc}$ | Short Circuit Current Density |
| LCOE | Levelized Cost of Energy |
| LT | Low Temperature |
| LT-CPSC | Low Temperature Carbon Perovskite Solar Cell |
| MA | Methylamine |
| MAI | Methylammonium Iodide |

| | |
|---|---|
| MC | Mesoporous Carbon |
| NiO | Nickel Oxide |
| NREL | National Renewable Energy Laboratory |
| OPV | Organic Photovoltaic |
| OSC | Organic Solar Cell |
| P3HT | poly(3-hexylthiophene-2,5-diyl) |
| Pb | Lead (element) |
| PCE | Power Conversion Efficiency |
| PHJ | Planar HeteroJunction |
| Pm | Maximum Power |
| PSC | Perovskite Solar Cell |
| PSM | Perovskite Solar Module |
| Pt | Platinum (element) |
| PV | Photovoltaic |
| QD | Quantum Dot |
| QD-SSC | Quantum Dot Sensitized Solar Cell |
| RH | Relative Humidity |
| Se | Selenium (element) |
| SEM | Scanning Electron Microscopy |
| Spiro-OMeTAD | 2,2′,7,7′-Tetrakis[N,N-di(4-methoxyphenyl)amino]-9,9′-spirobifluorene |
| SWCNH | Single Wall Carbon Nanohorn |
| SWCNT | Single Wall Carbon Nanotube |
| TCO | Transparent Conductive Oxide |
| TEM | Transmission Electron Microscopy |
| Ti | Titanium (element) |
| $TiO_2$ | Titanium Dioxide |
| UV | Ultraviolet |
| US | United States |
| $V_{oc}$ | Open Circuit Voltage |
| Zn | Zinc (element) |
| $ZrO_2$ | Zirconium Dioxide |

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
