# Peer review of "Carbon Electrodes: The Rising Star for PSC Commercialization"

_electronics, doi:10.3390/electronics12040992_

Round 1
Reviewer 1 Report
The authors' review is very informative, comprehensive_ extensive and deserves publication. The main drawback of the review is that it contains a lot of abbreviations and symbols. Moreover, many of them are used without decryption and explanation, while others are entered many times in different places of the text (up to three times). This makes it difficult to perceive the text. For the convenience of reading the review, it is extremely necessary to compile a list of abbreviations and physical symbols used. It is also necessary to refine the style of quoting literature (see below) and captions to drawings reproduced from other articles. A detailed list of items requiring adjustments is given in the file attached.

Author Response
We thank the Reviewer for his/her careful and thorough consideration of our work and the positive feedback. The comments have been addressed and have helped us improve the quality of this manuscript.
A Table of abbreviations and symbols has been added to the manuscript as Supplementary Information (SI). All References have been corrected as proposed. Excessive use of abbreviations has been addressed, as well as the decryption of abbreviations that was lacking, throughout the text. All captions of Figures have been made identical.
An explanatory sentence has also been added to define the connection between the aperture area, the active area and the dead area, which are directly connected to the calculation formula of the geometric fill factor.
Reviewer 2 Report
The manuscript –“Carbon electrode: the rising star for PSCs commercialization”– reviews the potential of high performing perovskite solar modules (PSMs) with carbon (C) electrodes which are prepared by low cost starting materials, using simple, scalable and industrially compatible methods. An overview of the background and current status, as well as the challenges and future prospects are also presented.
Due to the low cost and the large tunability in the chemical functionality and morphology, carbon is always the most promising candidate to replace precious metals in materials that require electronic conductivity, such as electrodes. On the other hand, the relationships between their functionality and the structure, composition, fabrication process are sometime very difficult to understand. Even under these circumstances, the focus of this review is very clear. Except for some parts, the authors describe the significance of carbon electrodes for perovskite solar cells, the fabrication method, the structure of the device including it, factors that can affect its performance, problems, and future prospects in a logical and easy-to-understand manner. Overall, this is a nice piece of work, and should definitely be reported. In the following, I raise a piece of comments to improve the paper before publication.
Comment1: For the Background section, the authors review the various carbon materials with different nanostructures, such as carbon black, CNTs, CNFs, CNHs, mesoporous carbon, single-walled CNH, CNTf, CTNCs, AnCs used in solar cell field (particularly in DSSCs). However, it is unclear why such nanostructures are necessary and can improve the solar cell performance. A brief description of each carbon material would be helpful to the reader. In fact, I think that those nanostructures in carbon materials are not so important for the main topic, that is, perovskite solar cell.
Comment2: The Current status – state of the art section seems to just list the results or track historical developments. Even though Table 3 is useful, I think this section needs a clear logical structure and corresponding subsections.
Author Response
We thank the Reviewer for the positive feedback on our manuscript.
Comment 1: The scope of mentioning the carbon nanostructures that have been used in other solar cell technologies than perovskite solar cells is only to predispose the readers on the potential of the material in solar cell technology in general. The appropriateness of C as the back electrode is demonstrated by the positive outcomes of its use in a variety of solar cells in the past, as well as the present. The description has been kept to minimum in order to present some of the highest performing materials, so the reader can look up the detailed works that have been presented, but without digressing from the main topic which is perovskite solar modules. We believe that the addition of details on carbon nanostructures that have been used in DSSCs will be out of the scope of this study.
Comment 2: We agree with the Reviewer and the text of "Current status-state of the art" part has been reorganized and divided into subsections, namely
4.3.1 Fully printable HT-CPSMs, 4.3.2 C electrode optimization, 4.3.3 Device optimization, 4.3.4 Stability, 4.3.5 LT-CPSMs
Reviewer 3 Report
This is an excellent review article prepared by the author. I have a few minor comments as stated below:
1- In some of the figures, the writing in the figure is not clear. Example, figure 1, figure 10, figure 28.
2- page 16, missing "e) spin coating" ?
3- for figure 12, the techniques stated in the figure, some of it does not link to the subtopic a), b), c), d) and e) stated above it.
4- for figure 28, the citation style should be in numbers.
5- English needs minor improvement.
Author Response
We thank the Reviewer for his/her positive feedback on our manuscript. Below there is the list of comments that have been addressed after his/her suggestions.
- We thank the reviewer for his/her constructive comment. The style of some figure captions presentation like those the reviewer is referred, strictly follows the directions of the publisher that holds the rights of them. As it concerns the size/analysis of them, we were obliged to follow the prototype figure hoping that it will be improved to the final published version of our manuscript
- In this part of the manuscript we have intentionally avoided to devote a separate section for the spin-coating process, since it is not commonly applied to large area PSMs, in fact it is a method to be avoided, therefore we have only made a very brief report.
- Figure 12 has been inserted as a summary Figure of several techniques that can be used for the fabrication of C electrodes, that are intended for use in PSCs. However, not all of them can be applied to the large area. The methods that are described in the subtopics a-d are the methods that are being applied in large area PSMs, which is the focus of this manuscript.
- The reason for this inconsistency with the other citation styles used for the Figures is that, contrary to the Publishers of the rest of Figures reproduced in this Review, the Publisher that owns the rights and permissions to use the particular figure requires this specific citation style.
- Minor English improvements have been performed.
Reviewer 4 Report
Review article on various studies Carbon (C) electrodes applied in photovoltaive cells.
Despite the interest that this technology may have and the review effort made by the authors, from the reviewer's point of view, the article lacks scientific interest in terms of providing clear scientific novelty. The Abstract does not clearly define the scientific novelty or usefulness of the review work.
The introduction does not clearly establish the main or secondary objectives of the work that justify it. (quality, economy, useful life, etc...)
The Review Methodology does not appear and the conclusions are only a list of very varied studies without a logical classification.
The sources of the figures are not clear and the figures are of very low quality so they cannot be analyzed clearly.
The tables are not classified, nor do they use the format established by the journal. There are no referenced studies that justify, from the technical, economic or sustainability point of view, the change in technology for the proposal.
There is no discussion of the bibliographic review carried out , a fundamental aspect in this type of work .
The Conclusions are very predictable and do not contribute scientific novelty.
It is recommended to rewrite the article.
Author Response
We thank the Reviewer for taking the time to read our manuscript.
Considering that the comments that were provided by the Reviewer lack a scientific base and point of view, that no details related to the topic of the manuscript (Carbon Perovskite Solar Modules) have been provided to justify the objection of the Reviewer and that no logical suggestions have been made in order to improve the quality of the work, that is in line with the Journal's guidelines, we would like to assume that the particular review of this manuscript was an unfortunate event on behalf of the Reviewer and not a targeted and intentional rejection, which serves some purpose.
Our review article has a clear direction to the module construction of third generation photovoltaics based on feasible methods of preparation, such as carbon based cathode materials, instead of the non feasible solution of gold evaporation. The scientific novelties that have contributed to this field are presented in this study. The general comments such as the excellence regarding quality, economy, useful life, etc., that is commented by the Reviewer, are in fact addressed in the manuscript, through the highlighting of the advantages of C-PSMs over typical PSMs. Any further analysis is out of the purpose of the specific work.
Regarding the proper use of English, the authors hold Certificate of Proficiency in English, by the University of Cambridge, which is at the disposal of the Reviewer and/or Editor upon request.
Round 2
Reviewer 1 Report
The authors made the required corrections. The article is recommended for acceptance for publication (please correct CNTf-CE to FSCNF-CE only in line 113).
Author Response
We thank the reviewer for the very careful examination of our manuscript.
Definitely, we have made the correction in line 113 of the manuscript as he/she suggested